# DenseMarks: Learning Canonical Embeddings for Human Heads Images via Point Tracks

**Dmitrii Pozdeev[1], Alexey Artemov[1], Ananta R. Bhattarai[2], Artem Sevastopolsky[1]**

[1]Technical University of Munich (TUM)    [2]University of Bielefeld

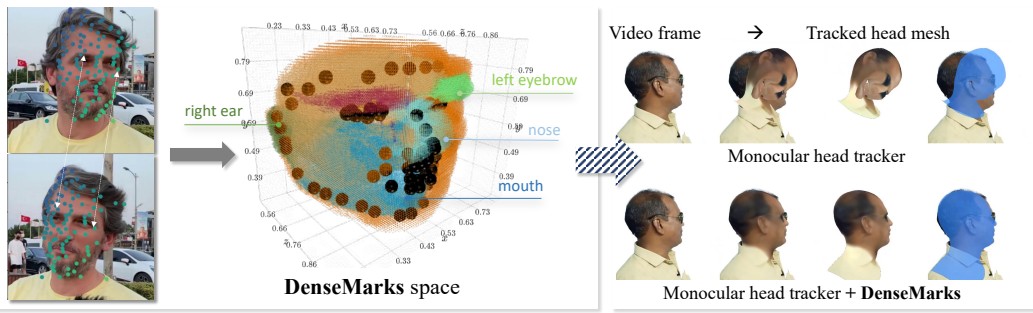

Figure 1: Our method learns to embed a human head image into a semantics-aware volumetric representation based on a large collection of in-the-wild talking head videos annotated by an off-the-shelf point tracker *(left)*. The embeddings can be estimated in a feedforward way and used for downstream applications, such as monocular tracking *(right)*, stereo reconstruction, and many others.

## Abstract

We propose DenseMarks – a new learned representation for human heads, enabling high-quality dense correspondences of human head images. For a 2D image of a human head, a Vision Transformer network predicts a 3D embedding for each pixel, which corresponds to a location in a 3D canonical unit cube. In order to train our network, we collect a dataset of pairwise point matches, estimated by a state-of-the-art point tracker over a collection of diverse in-the-wild talking heads videos, and guide the mapping via a contrastive loss, encouraging matched points to have close embeddings. We further employ multi-task learning with face landmarks and segmentation constraints, as well as imposing spatial continuity of embeddings through latent cube features, which results in an interpretable and queryable canonical space. The representation can be used for finding common semantic parts, face/head tracking, and stereo reconstruction. Due to the strong supervision, our method is robust to pose variations and covers the entire head, including hair. Additionally, the canonical space bottleneck makes sure the obtained representations are consistent across diverse poses and individuals. We demonstrate state-of-the-art results in geometry-aware point matching and monocular head tracking with 3D Morphable Models. The code and the model checkpoint will be made available to the public.

## 1 Introduction

Modern applications in augmented and virtual reality (AR/VR), telecommunications, computer gaming, and movie production require building models of humans at an increasingly demanding level of quality. Flaws in face and head modeling are particularly noticeable to human users (Haxby et al., 2000; Blanz & Vetter, 1999), so most human head modeling pipelines rely on head tracking (Thies et al., 2016; Giebenhain et al., 2024; Qian et al., 2024) to identify and maintain correspondences between head feature locations. Existing methods excel at features with consistent

statistical regularities across subjects. For instance, sparse facial landmark tracking (Lugaresi et al., 2019a; Cao et al., 2019) aims to follow the locations of unambiguous but isolated facial features shared by typical faces, such as the outlines of the eyes, nose line, or mouth corners. Similarly, parametric 3D model estimation and tracking (Blanz & Vetter, 1999; Li et al., 2017b; Dai et al., 2020) assumes that most face and head geometries follow a statistical shape model and can be represented by a shared, comparatively simple mesh template.

However, features like hair, accessories, and clothing are often omitted from head tracking, which typically focuses on landmarks or skin. In a typical video capture, individual landmarks or entire head regions easily become occluded due to an extreme pose, expression, or a worn accessory, introducing large errors in tracking. As a result, head tracks produced by conventional approaches are fundamentally limited by their incompleteness and correspondence instability.

To improve robustness of correspondence search, one path forward is to extract and match representations densely in each image pixel instead of detection and alignment of isolated landmarks. Recent image-based vision foundational models (VFMs) are one suitable source of such dense representations known to be effective in many vision tasks (Dutt et al., 2024; Siméoni et al., 2025). As human heads constitute a visual category with high structural similarity across instances, it is natural to expect such representations defined at unambiguous facial features to be nearly view- and time-invariant, facilitating exact correspondence search.

Building on these insights, we propose DenseMarks, a new learned representation for human heads designed to (1) enable high-quality dense correspondences for complete human heads, including irregular features such as hair or accessories, (2) achieve robust tracking under challenging conditions such as strong occlusions, and (3) produce a structured, interpretable, and smooth canonical latent space for exploration and interaction. We use a ViT neural backbone to predict dense per-pixel representations within the head mask of an input image; leveraging powerful pre-trained VFMs (Siméoni et al., 2025). These representations are projected into a shared 3D space, reducing correspondence to nearest-neighbor search and enabling intuitive interactions (e.g., click-based retrieval). To train without ground-truth dense correspondences, we construct a diverse dataset of human head videos with 2D point tracks from an off-the-shelf tracker (Karaev et al., 2024a). We enforce fine-grained cross-subject consistency by optimizing a contrastive loss on matched pairs, and integrate semantic and smoothness constraints to structure the latent space and improve interpretability.

We benchmark against pre-trained VFM variants (Siméoni et al., 2025; Khirodkar et al., 2024; Yue et al., 2024), with assessment focused on dense image warping and geometric consistency measures.

## 2 RELATED WORK

**Face, Head, and Full Body Tracking.** Commonly, tracking humans in videos involves extracting relevant information for the estimation and alignment of their pose and shape. In the simplest form, this is achieved by predicting locations of characteristic landmark points with fixed semantics (Sagonas et al., 2013; Moon et al., 2020; Jin et al., 2020) using learned models (Bulat & Tzimiropoulos, 2017; Lugaresi et al., 2019a; Cao et al., 2019; Simon et al., 2017; Li et al., 2022). Ease of collecting annotations and efficiency of landmark detectors have made landmarks essential in practical tracker design, enabling initial rigid alignment (Qian, 2024; Qian et al., 2024; Grassal et al., 2021; Bogo et al., 2016; Kanazawa et al., 2018; Kocabas et al., 2020). However, relying on a finite number of isolated, sparse landmarks can compromise robustness, commonly requiring regularization or postprocessing such as temporal smoothing (Qian, 2024; Zielonka et al., 2022; Zheng et al., 2023a; Huang et al., 2022; Jiang et al., 2022).

Many methods for estimating and tracking parametric models of faces and bodies (3DMMs (Blanz & Vetter, 1999; Zhu et al., 2017; Li et al., 2017b; Zhang et al., 2023b; Romero et al., 2017; Loper et al., 2015; Dai et al., 2020)) are based on the *analysis-by-synthesis* paradigm (Blanz & Vetter, 1999; Zhu et al., 2017; Feng et al., 2021; Zielonka et al., 2022; Daněček et al., 2022) that involves a combination of rigid alignment and optimization of denser losses. While offering higher geometric completeness, such models rely on a simple mesh topology and a limited range of geometries captured by a PCA basis (Abdi & Williams, 2010; Jolliffe, 2011); for fitting, they commonly depend on prior landmarks estimation and optimize highly non-convex (e.g., photometric or depth) losses.

Our method naturally complements 3DMM-based head trackers by supplying dense, robust semantic correspondences for complete heads and includes features not trivially captured by landmarks or parametric models (e.g., hair). This idea is similar to works that learn to predict texture coordinates for alignment of parametric face (Feng et al., 2018; Giebenhain et al., 2025) and body (Güler et al., 2018; Ianina et al., 2022) models, or compute multi-dimensional features, normals, and depth using foundation models optimized for the human domain (Khirodkar et al., 2024).

**Canonical Space Learning.** Our method represents input samples by learned embeddings in a shared *(canonical)* space. The idea of using canonical representations for category-level object localization and pose estimation was pioneered by Normalized Object Coordinate Space (NOCS) (Wang et al., 2019) and subsequently extended to handle sparse views, lack of dense labels, or multiple categories (Min et al., 2023; Xu et al., 2024; Krishnan et al., 2024). However, directly learning NOCS representations for 3D heads is difficult as large collections of 3D models are absent in the human head domain.

Shape correspondence task can be formulated as a problem of finding a mapping between spaces of functions defined on shapes (Ovsjanikov et al., 2012; Rodolà et al., 2017). Existing methods applying such functional maps for finding full-body correspondences (Neverova et al., 2020b; Ianina et al., 2022) require fitting parametric 3D models for supervision. To enable modeling parts of human heads absent from parametric models, we opted not to use these in our training.

The idea of using canonical space is widespread in 3D-aware per-scene human fitting (Gafni et al., 2021; Park et al., 2021) and human generative modeling EG3D (Chan et al., 2022; Dong et al., 2023). Similarly, several works focus on producing unsupervised shape correspondences, in part based on functional maps (Halimi et al., 2019; Cao & Bernard, 2022; Cao et al., 2023; Liu et al., 2025).

**Embeddings from Foundation Models.** Recent progress in ViT-based VFMs (Caron et al., 2021; Oquab et al., 2023; Siméoni et al., 2025; Weinzaepfel et al., 2022; Dosovitskiy et al., 2020; Han et al., 2022) and evidence of their emerging understanding of 3D world (Zhang et al., 2024b; Sucar et al., 2025; Chen et al., 2025a) has fueled efforts to improve their 3D-awareness through fine-tuning (Yue et al., 2024; Zhang et al., 2024a). Similarly, directly training siamese ViT networks on pairs of stereo views has been shown to efficiently establish dense correspondences (Wang et al., 2024; Leroy et al., 2024; Smart et al., 2024; Chen et al., 2025b), when prompted with 2+ images.

Another class of VFMs, pre-trained diffusion models (e.g., Stable Diffusion (Rombach et al., 2021)), allow inferring semantic correspondences from their image-based representations (Hedlin et al., 2023; Zhang et al., 2023a; Zhu et al., 2024) that could be distilled into dense surface correspondences across objects of arbitrary categories (Dutt et al., 2024). In our experiments, we found the correspondences arising from point tracking (cf. next paragraph) more reliable than those arising from pretrained diffusion models. Our method benefits from integrating VFMs as a feature extractor; in contrast to generic pre-trained deep features correlated with visual semantics, our geometry-aware representations yield an interpretable 3D canonical space.

**Point Tracking.** The advent of talking heads datasets (Wang et al., 2021; Zhu et al., 2022; Ephrat et al., 2018) and point trackers calls for approaches to tracking faces and bodies, free of an underlying coarse parametric model. In particular, in a line of works starting from PIPs (Harley et al., 2022), deep learning based methods are proposed to track any queried point along the video. Progress in the area of point trackers has been additionally accelerated by the appearance of suitable benchmarks, such as Tap-Vid (Doersch et al., 2022) and PointOdyssey (Zheng et al., 2023b). A series of consequent improvements of track-any-point algorithms (Doersch et al., 2023; Li et al., 2024; Cho et al., 2024) led to the emerging branch of CoTracker works (Karaev et al., 2024b;a), as well as BootsTAP (Doersch et al., 2024). Similarly, a few methods rely on foundation models, such as DINO-tracker (Tumanyan et al., 2024) for tracking any point or VGGT (Wang et al., 2025) that uses point tracks for 3D understanding. Applications of modern algorithmic ideas for point tracking also led to the appearance of simultaneous reconstruction and tracking methods such as Dynamic 3D Gaussians (Luiten et al., 2024), St4rTrack (Feng et al., 2025), or Tracks-to-4D (Kasten et al., 2024). For the downstream tasks of human tracking, similar to our method, some of the recent approaches also make use of point tracking (Kim et al., 2025; Taubner et al., 2024) or motion data (Shin et al., 2024).

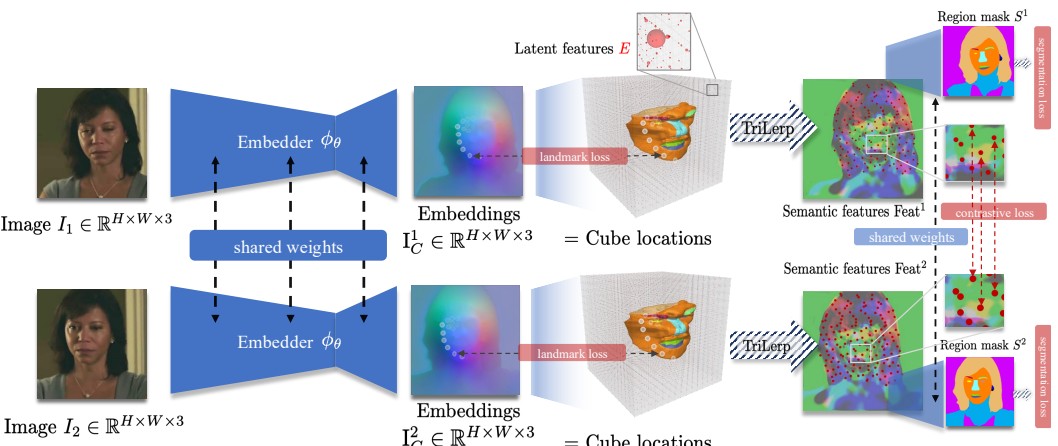

Figure 2: To learn our representation, we train an embedder network $\phi_\theta$ in a siamese fashion. By feeding two image frames from a talking head video of the same person into the embedder independently, we obtain DenseMarks embeddings $I_C^1, I_C^2$. These embeddings correspond to canonical locations in the unit cube (DenseMarks space). This cube is discretized in advance, and a learnable matrix $E$ of latent features represents $D$-dimensional vectors, storing semantic info of each of the voxel grid locations. To transform each of the estimated cube locations into semantic features $\text{Feat}^1, \text{Feat}^2$, we query $E$ at locations $I_C^1, I_C^2$ via trilinear interpolation (TriLerp). For the images $I_1, I_2$, we have a set of pair matches $K_{\text{gt}}^1, K_{\text{gt}}^2$, estimated by an off-the-shelf point tracker (Karaev et al., 2024a). We apply contrastive loss (Radford et al., 2021) to the semantic features of images in these locations. This way, the cube locations corresponding to the same semantic feature are pushed closer together. Additionally, we estimate region masks $S^1, S^2$ by a semantic network $S_\xi$ and apply segmentation loss.

## 3  METHOD

In this section, we define the representation (section 3.1) and the way a [2D image $\rightarrow$ embeddings] estimator is trained (section 3.2). The method overview is illustrated in Figure 2.

### 3.1  DENSEMARKS SPACE

The architecture of our pipeline consists of two key components: the canonical space where the embeddings reside, and the embedder, the task of which is to map an image into this space. The requirements that we set for the space are: (1) interpretable and queryable (the user can query a point in the space by looking at a typical arrangement of regions in it); (2) structured (regions are meaningful and don't overlap); (3) complete (contains the whole head, including the parts that are not trivial to annotate, such as hair and accessories); (4) smooth and continuous (images will be mapped to a continuous manifold of the space, with regions not getting abrupt and intersecting each other).

Additionally, we know that human heads are 3D objects. Even though UV (i.e., 2D canonical space) is a typical surface representation for heads, it's not the most precise representation due to modeling a complete head, including, e.g., hair and accessories, being not trivial in UV space and featuring seams (Ianina et al., 2022). Because of this, we decide to represent our canonical space as a unit cube in 3D and make the canonical embeddings the locations in this cube.

The interpretability requirement (1) and structure requirement (2) are enforced via landmark and segmentation losses, defined further in section 3.2.

The completeness requirement (3) is enforced with the way the embedder is supervised (also see section 3.2). For that purpose, we add a latent grid on top of the cube of a given resolution $N_d \times N_d \times N_d$ and attach a $D$-dimensional latent feature to each element of the voxel grid, thus forming a learnable matrix $E_{\text{raw}} \in \mathbb{R}^{(N_d)^3 \times D}$. Each latent feature contains a highly-dimensional info about the given location in the canonical space.

Finally, to promote the smoothness requirement (4), we apply spatial smoothness to the matrix $E_{\text{raw}}$ via a 3D Gaussian filter with a strength of $\sigma$, thus creating a latent feature grid $E = \text{gaussian\_filter\_3D}(E_{\text{raw}}, \sigma)$. This encourages the predicted embeddings from the embedder to be smoother, since the semantics of the close points in the cube will be similar and smoothly changing.

Note that the use of matrix E is inspired by the similar matrix of latent features used in functional maps, e.g., in CSE (Neverova et al., 2020b), that is typically smoothed via a Laplace-Beltrami operator (Lévy, 2006). From a different standpoint, the operation of querying the space can also be seen as an attention operation, where the locations are queries (same as keys in this context) and the latent grid features are values. By aggregating the values at real-valued query locations with trilinear interpolation weights, we obtain the resulting semantic features at a given location.

## 3.2 EMBEDDER TRAINING

Our goal is to learn a monocular embedder $\psi_\theta : \text{I} \to \text{I}_C$, where $\text{I} \in \mathbb{R}^{H \times W \times 3}$ is an input RGB image and $\text{I}_C \in \mathbb{R}^{H \times W \times 3}$ is the predicted canonical embeddings for each pixel.

The network consists of a Vision Transformer backbone that predicts a feature map, which is further gradually upscaled through a sequence of convolutional layers to match the input resolution.

To train this network, at each training step, we pass two input images $\text{I}^1, \text{I}^2 \in \mathbb{R}^{H \times W \times 3}$ through the embedder $\psi_\theta$ and obtain corresponding predictions $\text{I}_C^1 = \psi_\theta(\text{I}_1), \text{I}_C^2 = \psi_\theta(\text{I}_2)$, both in $\mathbb{R}^{H \times W \times 3}$. For these two images, we assume having a number of ground truth pixel correspondences between them $(K_{\text{gt}}^1, K_{\text{gt}}^2) = \left(\{(i_1^1, j_1^1), \ldots, (i_P^1, j_P^1)\}, \{(i_1^2, j_1^2), \ldots, (i_P^2, j_P^2)\}\right)$. These correspondences could be coming from any off-the-shelf pairwise matching algorithm. In our case, we obtain them from a point tracker inferred over individual talking head videos, as we found best in practice. Because of this, in our training procedure, images $I_1$ and $I_2$ are always coming from the same talking head video, but can represent arbitrarily close or far frames of the same video.

Embeddings $\text{I}_C^1 = \psi_\theta(\text{I}_1)$ and $\text{I}_C^2 = \psi_\theta(\text{I}_2)$ point to some real-valued locations in the canonical space. For each of those, we extract their corresponding $D$-dimensional semantic features via trilinear interpolation (Trilerp) (Bourke, 1999): $\text{I}_{\text{feat}}^1 \in \mathbb{R}^{H \times W \times D}, \text{I}_{\text{feat}}^2 \in \mathbb{R}^{H \times W \times D}$, where $(\text{I}_{\text{feat}}^1)_{ij} = \text{Trilerp}(E, (\text{I}_C^1)_{ij}), (\text{I}_{\text{feat}}^2)_{ij} = \text{Trilerp}(E, (\text{I}_C^2)_{ij})$.

In order to supervise our network, we encourage the features $\text{I}_{\text{feat}}^1, \text{I}_{\text{feat}}^2$ to be close at the positions, defined by ground truth correspondences $(K_{\text{gt}}^1, K_{\text{gt}}^2)$, and far for other pairs of points. More formally, we first extract semantic features at the integer spatial positions of the ground truth correspondences, yielding tensors of queried features $\text{Feat}^1, \text{Feat}^2 \in \mathbb{R}^{P \times D}$, $\text{Feat}_p^1 = \text{I}_{\text{feat}}^1[(K_{\text{gt}}^1)_p]$, $\text{Feat}_p^2 = \text{I}_{\text{feat}}^2[(K_{\text{gt}}^2)_p]$. To promote the corresponding features of the first and second image to be close (*positive pairs*) and the others to be far (*negative pairs*), we construct a contrastive loss similar to CLIP Loss (Radford et al., 2021) that requires the pairwise matrix of cosine distances to be close to an identity matrix:

$$\mathcal{L}_{\theta, E}^{\text{contr}}(\text{Feat}^1, \text{Feat}^2) = \left\| (\text{norm}(\text{Feat}^1))(\text{norm}(\text{Feat}^2))^T - I \right\|_F,$$

where *norm* is a row-wise normalization operation.

Additionally, we apply a number of regularizations. To reduce ambiguity of the learned canonical space, we impose the locations of standard 300W Sagonas et al. (2013) format face landmarks to be close to the predefined locations in the cube. This is implemented via inferring an off-the-shelf landmark predictor on images $\text{I}_1, \text{I}_2$, thus obtaining ground truth landmark locations $(l_1^1, \ldots, l_{68}^1), (l_1^2, \ldots, l_{68}^2)$, and anchoring them to the predefined locations $L_k \in \mathbb{R}^3$, $k = 1, \ldots, 68$ in the unit cube:

$$\mathcal{L}_\theta^{\text{lmks}}(\text{I}_C \mid \boldsymbol{l}) = \sum_{k=1}^{68} |\text{I}_C[l_k] - L_k|$$

To further correlate the predicted canonical embeddings with image semantics, we add a trainable segmentation head $\text{S}_\xi$, consisting of a single conv1x1 layer. For each of the images, this head receives the extracted semantic features (either $\text{Feat}^1$ or $\text{Feat}^2$) and returns the predicted logits of probabilities of class regions (face parsing) – either $S^1 = \text{S}_\xi(\text{Feat}^1)$, or $S^2 = \text{S}_\xi(\text{Feat}^2)$, both in $\mathbb{R}^{H \times W \times N_S}$. The segmentation loss expression compares each of the predicted masks $S \in \{S^1, S^2\}$

to the corresponding ground truth mask $S_{\text{gt}} \in \mathbb{R}^{H \times W \times N_S}$, obtained by an off-the-shelf face parser:

$$l^{\text{segm}}(S \mid S_{\text{gt}}) = \sum_{i,j} \text{cross\_entropy}(S[i,j], S_{\text{gt}}[i,j])$$

The overall loss is as follows:

$$
\begin{aligned}
\mathcal{L}_{\theta,E,\xi}(\cdot) = {} & \mathcal{L}_{\theta,E}^{\text{contr}}(\text{Feat}^1, \text{Feat}^2) \\
& + \lambda_{lmks}(l_\theta^{\text{lmks}}(\text{I}_C^1 \mid \boldsymbol{l}^1) + l_\theta^{\text{lmks}}(\text{I}_C^2 \mid \boldsymbol{l}^2)) \\
& + \lambda_{segm}(l^{\text{segm}}(S^1 \mid S_{\text{gt}}^1) + l^{\text{segm}}(S^2 \mid S_{\text{gt}}^2))
\end{aligned}
\tag{1}
$$

## 4 EXPERIMENTS

### 4.1 EXPERIMENTAL SETUP

**Data.** We train our method on CelebV-HQ dataset (Zhu et al., 2022) of 35K in-the-wild talking head videos of interview style. To obtain ground truth correspondences $(K_{\text{gt}}^1, K_{\text{gt}}^2)$, we run Co-Tracker3 (Karaev et al., 2024a) on these videos. As an input set of points to track, we take the whole foreground region of the first frame (estimated by GroundedSAM2 (Ren et al., 2024) prompted with the text *"person"*) and sample points uniformly in that region (see an example in Fig. 1 *(left)*). Videos were discarded if there were either too few tracks found (fewer than 80) or foreground segmentation failed, resulting in 32K videos left. The number of point tracks found did not exceed 400. We show samples of point tracks visualisation in the appendix J. 100 randomly sampled videos have been held out for the evaluation and used in the results described below. Each training batch is formed by uniformly sampling two random frames from a sample video from the constructed annotated dataset. All videos are resized to the (512, 512) resolution in advance and fed to the embedder in that resolution. For augmentation, we use random shift (in [-10%, 10%] range), scale ([-10%, 10%]), and rotation ([-18°, 18°]), each with a chance of 50%. Points which are no longer visible after augmentation are no longer accounted in training. For the landmark loss, we extract 70 manually selected landmarks (full face border, landmarks on eyes, nose, and mouth) via Mediapipe (Lugaresi et al., 2019b). Ground truth segmentation masks are obtained via FaRL (Zheng et al., 2022b) and are further refined on the borders via face-parsing (Jonathan Dinu, 2025; Xie et al., 2021), which works better in practice on non-face regions of the head.

**Architecture and training.** To make use of strong pretraining, we initialize the embedder with a pre-trained DINOv3 (Siméoni et al., 2025) checkpoint and add DPT head (Ranftl et al., 2021) to output an image of the same spatial resolution as the input ($512 \times 512$). Matrix E is initialized from a Gaussian distribution $\mathcal{N}(0,1)$. We use $\lambda_{segm} = 1$ for the segmentation loss and $\lambda_{lmks} = 50$ for the landmark loss. For optimization, we employ the AdamW (Loshchilov, 2017) optimizer with a learning rate $5 \cdot 10^{-5}$ for the backbone of $\phi_\theta$, learning rate of $10^{-4}$ for DPT head, and $10^{-3}$ for the latent features $E$. The schedule for all learning rates was cosine annealing with an overall number of steps of 140K and a warmup for 2'800 steps. Weight decay of $10^{-4}$ was applied to the network parameters $\theta$ and $\xi$, except for normalization layers. The whole pipeline is trained for $140k$ training steps using 8 pairs of images per batch on a single NVIDIA RTX 3090 Ti GPU for 1.5 days.

**Baselines.** We compare against state-of-the-art general-purpose dense feature extractors, the embeddings of which provide rich semantic information: DINOv3 (Siméoni et al., 2025) (embedding dimension: 768), Diffusion Hyperfeatures (Luo et al., 2023) (384), Fit3D (Yue et al., 2024) (768). Additionally, we consider head-centric dense feature extractors: Sapiens (Khirodkar et al., 2024) (1280), CSE (Neverova et al., 2020a) (16) and FaRL (Zheng et al., 2022b) (768). All of our baselines predict dense feature embeddings that can be used directly for a neighbor / region search. For visualizations, we choose the two strongest baselines from each category. For completeness, we add comparison with 3DMM-related methods in the appendix I.

### 4.2 RESULTS

**Point querying.** The requirement of the canonical space is that the same semantic points will have a fixed location in the cube, regardless of the person's identity. We test this on a number of points that have distinct semantics: points on hair, ear centers, forehead center, eyebrow corners.

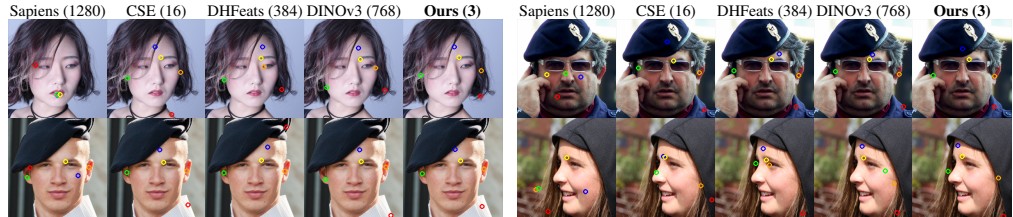

Figure 3: Point querying. We select a specific point on a few images and find the reference embedding by averaging the embeddings predicted by each of the models in its location. Points: red = on the left side of long hair region, green = center of the right ear, orange = center of the left ear, blue = forehead center, yellow = left eyebrow corner. We indicate the embedding dimension in brackets.

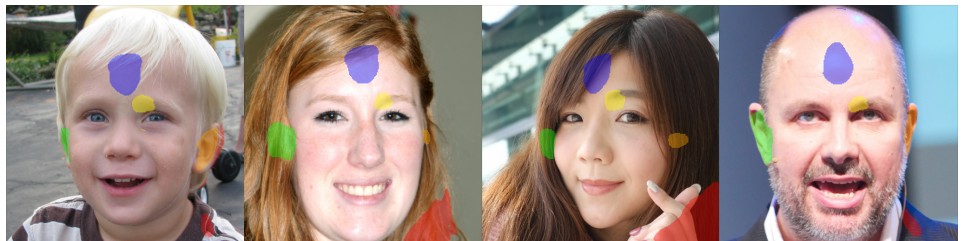

Figure 4: Semantic regions on head images can be located via selecting corresponding volumetric regions in the canonical space. Red: on the left side of long hair region, blue: forehead center, green and orange: ears, yellow: skin near the left eyebrow corner.

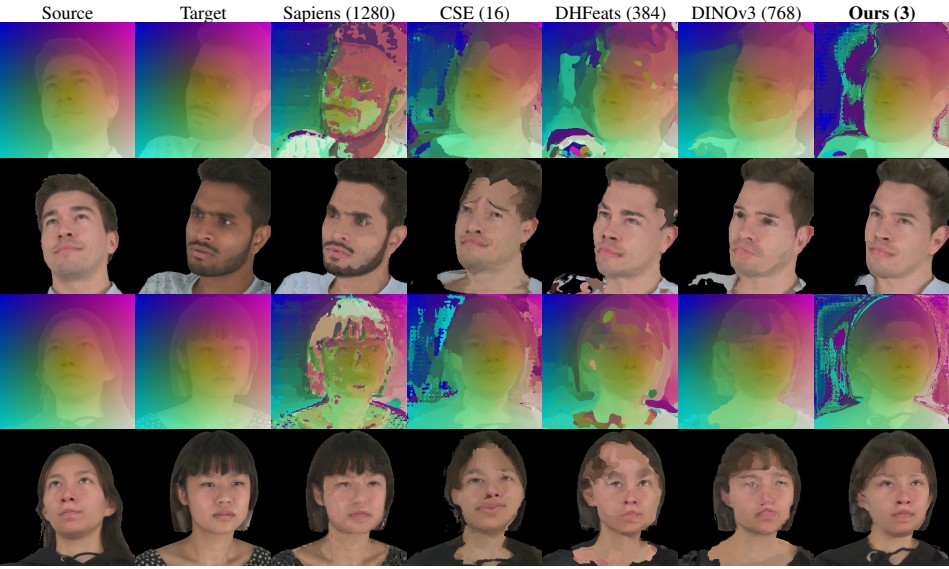

Figure 5: Dense warping. Here, we copy pixels from source to target based on the target→source nearest neighbors search in the space of embeddings, predicted by each model *(even rows)*. For clarity, mapping of meshgrid-like coordinates, blended with RGB, is shown additionally *(odd rows)*. Even though deep feature extractors provide valuable matches, they are either matching colors, not semantics (Sapiens (Khirodkar et al., 2024), DHFeats (Luo et al., 2023)), or feature significant artifacts (DINOv3 (Siméoni et al., 2025), CSE (Yue et al., 2024)), thus being less reliable for matching.

To find each semantic point, we manually annotated 7 sample images from CelebV-HQ, inferred the trained embedder, and averaged predicted locations in the cube for each annotated point. We use the obtained location as a reference to find the nearest neighbor in the other image among their predicted embeddings. Results are demonstrated in Fig. 3. For our baselines, semantic points are

Table 1: Quantitative comparison. On same-person pairs of images from Nersemble (Kirschstein et al., 2023), we evaluate the quality of correspondences that arise from matching nearest neighbor embeddings. Similarly, on cross-person pairs, we evaluate the consistency and identity preservation.

| | | Same-person | | | Cross-person | |
| --- | --- | --- | --- | --- | --- | --- |
| | | Matching quality | | | Identity preservation | |
| | | MAE ↓ | RMSE ↓ | PCK@r=0.05 ↑ | ArcFace ↑ | Met3R ↓ |
| Generic | Fit3D (Yue et al., 2024) | 12.75 | 21.83 | 0.57 | 0.236 | 0.558 |
| | Hyperfeatures (Luo et al., 2023) | 8.26 | 13.29 | 0.72 | 0.329 | 0.454 |
| | DINOv3 (Siméoni et al., 2025) | 7.60 | 12.69 | 0.72 | 0.266 | 0.460 |
| head-centric | FaRL (Zheng et al., 2022a) | 21.38 | 34.41 | 0.46 | 0.166 | 0.632 |
| | Sapiens (Khirodkar et al., 2024) | 14.88 | 24.12 | 0.56 | 0.167 | 0.595 |
| | CSE (Neverova et al., 2020b) | 11.22 | 17.92 | 0.55 | 0.359 | 0.490 |
| | Ours | **3.68** | **5.90** | **0.90** | **0.384** | **0.388** |

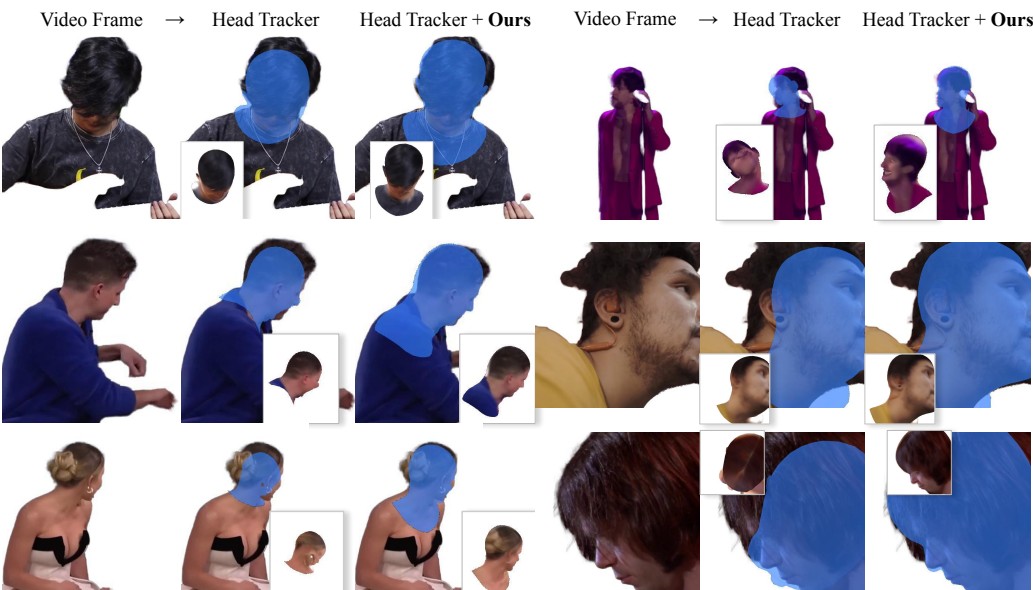

Figure 6: Monocular tracking. We evaluate our method on downstream application of applying a state-of-the-art off-the-shelf head tracker (Qian, 2024) to track a 3D Morphable Model template (FLAME (Li et al., 2017a)) over a monocular video. By default, this tracker relies on standard 68 face landmarks and photometric loss. Estimating a DenseMarks texture of FLAME and applying an additional photometric loss to match it with estimated embeddings greatly improves the robustness of the tracker, especially for extreme poses.

also estimated by averaging predicted embeddings. Despite using a significantly smaller vector dimension (3) to store semantics in the embedding, our method can find a corresponding region for challenging views better. Note that our method is also robust to strong face or head occlusions.

**Region selection.** In Fig. 4, we demonstrate how the same volumetric region in the canonical space is mapped onto images of people. The regions are initially selected on 7 random images manually and averaged (via a voting procedure) in the cube space.

**Dense warping.** To demonstrate the semantic consistency of embeddings predicted for the whole image, not only specific points or regions, we demonstrate the warping by embeddings in Fig. 5, evaluated on pairs of different

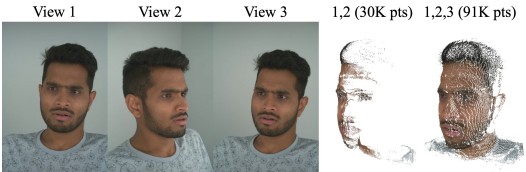

Figure 7: Stereo Reconstruction. We triangulate 2-view and 3-view correspondences of our representations using known camera parameters in Nersemble (Kirschstein et al., 2023).

Table 2: Importance of the canonical space. On same-person pairs of images from Nersemble (Kirschstein et al., 2023), we evaluate the quality of correspondences that arise from matching nearest neighbor embeddings. Similarly, on cross-person pairs, we evaluate the consistency and identity preservation.

| | MAE ↓ | RMSE ↓ | PCK@r=0.05 ↑ | ArcFace ↑ | Met3R ↓ |
|---|---|---|---|---|---|
| w/o canonical space | 6.35 | 10.20 | 0.85 | 0.348 | 0.455 |
| Ours | **3.68** | **5.90** | **0.90** | **0.384** | **0.388** |

people from the Nersemble dataset (Kirschstein et al., 2023). For each target image pixel, we replace its color with the color of the nearest neighbor by embedding in the source. We expect the warping to be semantically meaningful and smooth. It is observed that when we match nearest neighbors by Diffusion Hyperfeatures and especially Sapiens embeddings, the matches turn out to be based on the color similarity, not the semantic similarity. DINOv3 and CSE appear more semantically meaningful but often feature artifacts, making the correspondences imprecise, as best observed in the mapping rows in the figure. To evaluate the quality of the mapping, we estimate face recognition similarity based on ArcFace (Deng et al., 2019) between the source image and the mapping result, as well as the view-consistency metric Met3R (Asim et al., 2025), and show the results in Table 1.

**Geometric consistency.** To assess qualitatively and quantitatively the precision of the estimated correspondences through our embeddings, we repeat the Dense Warping experiment in a similar way for the (source, target) pairs of images of the same person, not different people, repeated over various people from the Nersemble dataset. In Table 1, we demonstrate the evaluation of the correctness of the estimated correspondences between source and target, averaged over ten people from Nersemble. As a source of ground truth correspondences, we estimate a complete head mesh from all 16 cameras via GS2Mesh (Wolf et al., 2024) and sample 1K random mesh vertices. The embeddings are evaluated in the projected locations of these vertices.

**Canonical space ablation.** To separate the importance of the canonical space from pure representation learning, we train our model without introducing the canonical space learned by our method. While training on head-specific data improves geometric and semantic consistency relative to DINOv3, our method still outperforms this approach (see Table 2 and App. D).

**Design choices ablation.** Even though the network can learn without introduced constraints on landmark locations in the cube and segmentation loss, we demonstrate that finding characteristic points and regions becomes more problematic in Fig. 8. This is explained by a less semantically constrained canonical space. We provide quantitative analysis of $\lambda_{lmks}, \lambda_{seg}$, different smoothing strategies for $E_{raw}$ and effect of cube resolution $N$ in App. B.

**Robustness to pseudo-GT tools.** The quality of DenseMarks depends on the robustness of the tools used to generate ground-truth labels. Errors in point-tracks arise mainly from (1) missing tracks (most common) and (2) inaccurate track positions. To simulate this, for (1) we randomly retain only a fraction of visible tracks; for (2) we add Gaussian noise with increasing deviation (pixels) to track locations. Results are shown in Table 3. Although stronger noise degrades accuracy more than track omission, even with severe noise (±16 px), the method still outperforms all baselines. We observe even stronger robustness of our method against errors in landmarks and seg-

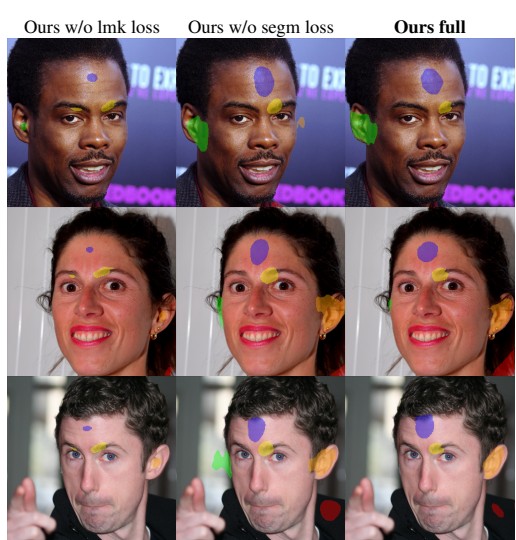

Figure 8: Removing the landmark or segmentation loss makes region finding much less reliable. Blue: forehead center, green and orange: ears, yellow: skin near the left eyebrow corner.

mentation masks (see App. C). Overall, these results indicate that the pseudo-GT tools provide reliable supervision and that the learned representations are resilient to moderate noise. This suggests

Table 3: Stability of Densemarks to the errors of the point-trackers. (Left): adding gaussian noise to the tracks with standard deviation $\sigma$. (Right): keeping only a fraction of the original tracks. Our method remains effective under mild noise. We use 50% of the training steps to reduce computational costs.

| Method | MAE ↓ | RMSE ↓ | ArcFace ↑ | Met3R ↓ | Method | MAE ↓ | RMSE ↓ | ArcFace ↑ | Met3R ↓ |
|---|---|---|---|---|---|---|---|---|---|
| Keep 10% | 4.52 | 7.30 | 0.376 | 0.391 | $\sigma = 16$ | 6.29 | 9.97 | 0.355 | 0.436 |
| Keep 20% | 4.17 | 6.70 | 0.378 | 0.389 | $\sigma = 8$ | 5.14 | 8.20 | 0.361 | 0.411 |
| Keep 40% | 3.87 | 6.18 | 0.379 | 0.384 | $\sigma = 4$ | 4.26 | 6.81 | 0.382 | 0.390 |
| Keep 80% | 3.86 | 6.19 | 0.381 | 0.383 | $\sigma = 2$ | 3.86 | 6.19 | 0.383 | 0.378 |
| Ours (100%) | 3.85 | 6.17 | 0.388 | 0.384 | Ours ($\sigma = 0$) | 3.85 | 6.17 | 0.388 | 0.384 |

DenseMarks may generalize to domains where accurate tracks are difficult to obtain, such as full-body capture or highly dynamic non-rigid objects.

**Monocular tracking.** As an example application of our method, we take a highly-performing off-the-shelf head tracker, VHAP (Qian, 2024), which supports estimation of the FLAME parametric head model (Li et al., 2017a). It relies on a standard 300-W set of 68 sparse landmarks (Sagonas et al., 2013) for rigid alignment of the template and optimizes for the shape, pose, and expression parameters of FLAME, through estimating RGB texture in the FLAME UV space and applying photometric loss. Even though VHAP excels in multi-view settings, monocular videos can remain challenging due to potentially failing landmark detection, occlusions, and extreme viewpoints. To aid the tracker in these situations, we add another photometric loss that is based on estimating a 3-dimensional UV texture of DenseMarks embeddings that is compared to the embeddings predicted by the trained embedder for each video frame independently. We run tracking on in-the-wild monocular videos with different challenging conditions such as strong/fast head rotation, severe hair/accessories occlusions, very close/far cameras. The results are demonstrated in Figure 6. Our method improves robustness the most in cases of extreme poses and yields better alignment in challenging regions, such as neck and ears. We demonstrate the results of tracking over the complete videos in the Supplementary Video.

**Stereo Reconstruction.** In Fig. 7, we demonstrate that triangulating 2+ images can be done purely using embeddings from our model, on the example of a sample from Nersemble with known camera poses and intrinsics. This way, we show the capabilities of [multi-view-]stereo and dense estimation.

**Other applications.** We additionally provide results on head pose estimation (App. F) and other objects (App. E). In App. G, we demonstrate robustness to strong lightning change, motion blur, color shifts, and synthetic occlusions.

**Limitations.** Our method relies on video human collection datasets featuring a wide range of appearances and containing videos of sufficient length and resolution of the head region. While CelebV-HQ is among the most extensive publicly available datasets (35k videos), it primarily consists of interview-style / film shooting data. As extremely challenging sequences, such as those featuring backside head views, extreme head rotations, and very rapid motion, are uncommon in this dataset, we expect a quality drop of DenseMarks on those sequences, especially on the monocular tracking benchmark, which requires most of the per-pixel correspondences to be reliable. We provide failure cases of monocular tracking and dense warping in the Supplementary Video and in the Appendix H.

## 5 CONCLUSION

We propose a novel representation for human head images and an embedder for dense estimation. The resulting low-dimensional (3D) embeddings are consistent across views and subjects, enabling reliable matching of challenging regions like hair. Despite their compactness, they outperform high-dimensional features from foundation models in geometry-aware tasks like tracking, while benefiting from VFM pretraining. Future work could extend our approach to full bodies and other domains, which would be anticipated with the appearance of publicly available high-resolution data collections.

## ACKNOWLEDGMENTS

Dmitrii Pozdeev is grateful to Freunde der TUM e.V. for the financial support to attend the conference. We are thankful to Visual Computing & Artificial Intelligence lab at TUM for providing the computing resources during Guided Research. We thank Shenhan Qian and Yash Kant for useful discussions. Alexey Artemov and Artem Sevastopolsky currently work at Apple. This paper is not connected to their work at Apple.

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

# A APPENDIX

**Use of LLMs.** We used LLMs for expanding our knowledge regarding the latest related work.

# B ABLATION STUDIES

To avoid computational burden, we use 50% of the available training steps, which, in our experience, is usually sufficient for decent convergence.

## B.1 LOSS WEIGHTS

Table 4: Ablation study on loss weights $\lambda_{lmks}$ and $\lambda_{seg}$.

| Method | MAE $\downarrow$ | RMSE $\downarrow$ | ArcFace $\uparrow$ | Met3R $\downarrow$ |
|---|---|---|---|---|
| Ours ($\lambda_{lmks} = 50, \lambda_{seg} = 1$) | 3.85 | 6.17 | 0.388 | 0.384 |
| $\lambda_{lmks} = 10$ | 5.35 | 8.68 | 0.390 | 0.413 |
| $\lambda_{lmks} = 250$ | 3.92 | 6.29 | 0.393 | 0.387 |
| $\lambda_{seg} = 0$ | 3.68 | 5.86 | 0.353 | 0.420 |
| $\lambda_{seg} = 0.2$ | 3.72 | 5.95 | 0.400 | 0.374 |
| $\lambda_{seg} = 5$ | 3.98 | 6.39 | 0.384 | 0.400 |

We provide quantitative ablation study on loss weights $\lambda_{lmks}$ and $\lambda_{seg}$ in Table 4. We observe that increasing $\lambda_{lmks}$ does not further improve the metrics, and making it too low ($\lambda_{lmks} = 10$) results in inferior model performance. For the segmentation loss weight, lowering $\lambda_{seg}$ to 0.2 yields slightly better performance on both single-person and cross-person benchmarks. As shown in Fig. 9, our model exhibits a close canonical space structure to the $\lambda_{seg} = 0.2$ baseline. However, removing the segmentation loss entirely ($\lambda_{seg} = 0$) leads to degraded cross-person metrics, indicating the importance of semantic supervision.

## B.2 SMOOTHING STRATEGIES

We compare different smoothing strategies applied to the latent feature grid $E$ in Table 5. We evaluate four approaches: (1) no smoothing; (2) uniform smoothing with the same kernel size (7×7); (3) a 3D bilateral filter with $\sigma_d = 1.5$ (spatial difference) and $\sigma_r = 0.1$ (intensity difference), selected from a grid search over [0.05, 0.1, 0.2, 0.4]; (4) Gaussian smoothing (our method) with $\sigma = 1.5$.

Overall, the quantitative evaluation confirms the importance of using a smooth canonical space, and the specific filter to use appears not to be particularly important.

Without smoothing, the model is still able to roughly locate correct regions in the canonical space, but the space is harder to use directly for querying or region finding, as evidenced by the less consistent region boundaries (see Fig. 10).

## B.3 GRID RESOLUTION AND SMOOTHING STRENGTH

We analyze the effect of grid resolution $N_d$ (the number of voxels in the voxel grid by each side) and Gaussian smoothing strength $\sigma$ in Table 6. We observe a slight improvement in geometric metrics (MAE, RMSE) with a lower sigma value ($\sigma = 0.5$). However, the best variant according to cross-person metrics (ArcFace, Met3R) is achieved with $\sigma = 1.5$, either with $N_d = 64$ for ArcFace or with $N_d = 128$ for Met3R (as in our final method). Overall, the metrics do not seem to deviate significantly with variations in these parameters.

**Memory consumption.** Memory occupied by the voxel grid grows cubically with $N_d$. The method takes approximately 16 GB GPU memory for $N_d = 128$ and 14 GB for $N_d = 64$. Increasing $N_d$ further significantly increases memory requirements to approximately 32 GB. Sigma does not affect memory consumption. We were unable to run experiments with $N_d = 256$ due to constraints on available GPU memory.

| Ours | Ours w. 0.2x $\lambda_{seg}$ | Ours | Ours w. 0.2x $\lambda_{seg}$ |

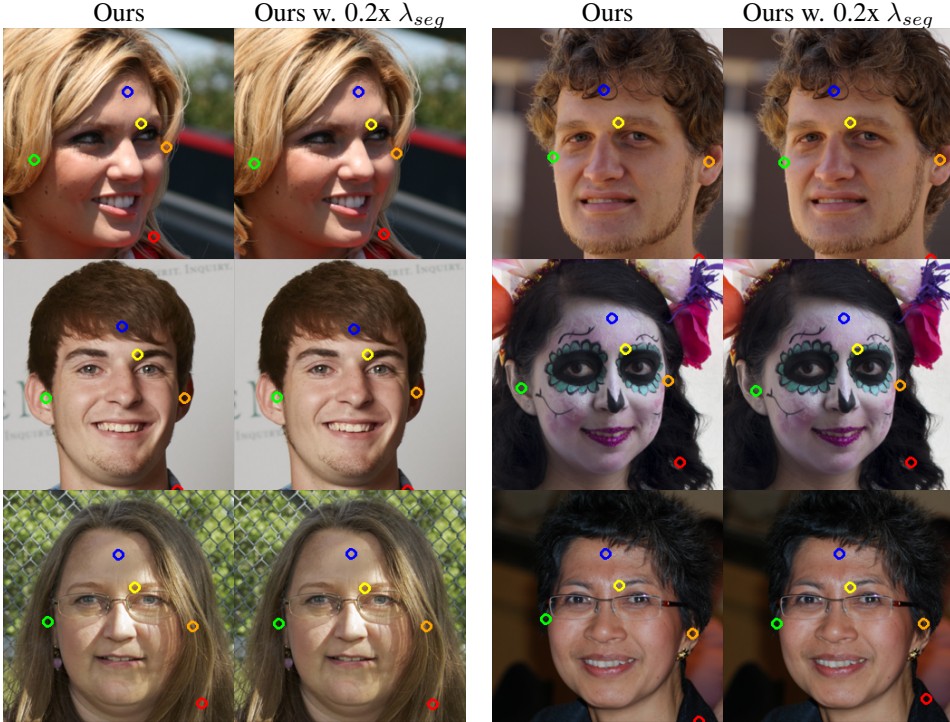

Figure 9: Semantic regions on head images can be located via selecting corresponding volumetric regions in the canonical space. Blue: forehead center, green and orange: ears, yellow: skin near the left eyebrow corner. Reducing segmentation loss weight $\lambda_{seg}$ results in a similar structure of the canonical space.

Table 5: Ablation study on different smoothing strategies.

| Method | MAE ↓ | RMSE ↓ | ArcFace ↑ | Met3R ↓ |
|---|---|---|---|---|
| No smoothing | 4.22 | 6.71 | 0.387 | 0.393 |
| Uniform smoothing | 3.83 | 6.14 | 0.383 | 0.381 |
| Bilateral filter | 3.97 | 6.35 | 0.392 | 0.390 |
| Gaussian smoothing (Ours) | 3.85 | 6.17 | 0.388 | 0.384 |

## B.4 POINT-TRACKER ABLATION

By default, CoTracker filters points by discarding them if their estimated confidence is below a threshold of 0.6; this threshold was also used in our experiments. Here, we demonstrate what happens when stricter filtering with a larger threshold is applied, thereby removing more outliers. Results are shown in Table 7.

In line with our observations on the stability of the point tracker, increasing the threshold mildly improves the results, as it removes outliers. With a stronger threshold of 0.75 and 0.9, we obtain performance similar to that of 0.6, which is used in our model. With a very strict threshold of 0.98, the model's performance slightly improves indeed. However, if we increase the threshold further (0.99), too many non-outliers are excessively filtered out, and no further improvement is observed.

Finally, we observe a slight performance boost when a more recent point-tracker is used for data acquisition (see Table 8). For fair comparison, we use the same initializations for point tracks as in Cotracker3.

Ours  No smoothing   Ours  No smoothing

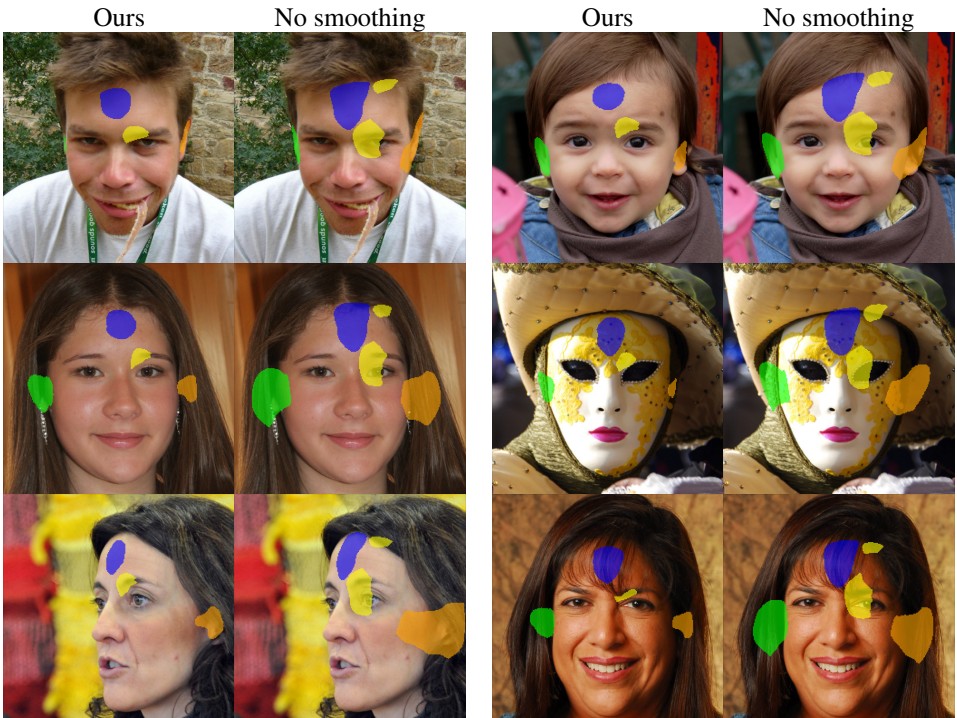

Figure 10: Semantic regions on head images can be located via selecting corresponding volumetric regions in the canonical space. Blue: forehead center, green and orange: ears, yellow: skin near the left eyebrow corner. Adding smoothing makes region finding more reliable.

Table 6: Ablation study on grid resolution $N_d$ and Gaussian smoothing strength $\sigma$.

| Method | MAE $\downarrow$ | RMSE $\downarrow$ | ArcFace $\uparrow$ | Met3R $\downarrow$ |
|---|---|---|---|---|
| $N_d = 64$, $\sigma = 0.5$ | 3.65 | 5.86 | 0.391 | 0.390 |
| $N_d = 64$, $\sigma = 1.5$ | 4.06 | 6.51 | 0.392 | 0.385 |
| $N_d = 64$, $\sigma = 4.5$ | 4.04 | 6.49 | 0.391 | 0.385 |
| $N_d = 128$ (ours), $\sigma = 0.5$ | 3.64 | 5.80 | 0.390 | 0.386 |
| $N_d = 128$ (ours), $\sigma = 1.5$ (ours) | 3.85 | 6.17 | 0.388 | 0.384 |
| $N_d = 128$ (ours), $\sigma = 4.5$ | 3.89 | 6.25 | 0.386 | 0.386 |

## C Robustness to Landmarks and Segmentation Masks

We further evaluate robustness to errors in landmarks and segmentation masks by adding Gaussian noise to landmarks and dilating randomly selected segmentation classes using a $3 \times 3$ kernel. Dense-Marks remains robust to large mask dilations (48 px) and strong landmark noise (up to $\pm 16$ px), showing greater tolerance to these errors than to track inaccuracies. Results are shown in Table 9.

## D Importance of the Canonical Space, Dense Warping

We provide additional results ablating the use of the canonical space in our method on dense warping in Fig. 11. Using a DenseMarks space provides better geometric awareness, as evidenced by the better consistency of the blended images.

## E Experiments on Chairs

To demonstrate the applicability of our method to other domains beyond human heads, we retrained our approach on a sample category of generic objects, specifically, chairs from the ShapeNet (Chang

Table 7: Effect of different confidence thresholds for filtering CoTracker points.

| Method | MAE ↓ | RMSE ↓ | ArcFace ↑ | Met3R ↓ |
|---|---|---|---|---|
| Confidence threshold = 0.6 (ours) | 3.85 | 6.17 | 0.388 | 0.384 |
| Confidence threshold = 0.75 | 3.76 | 6.00 | 0.384 | 0.386 |
| Confidence threshold = 0.9 | 3.81 | 6.11 | 0.385 | 0.389 |
| Confidence threshold = 0.98 | 3.63 | 5.81 | 0.390 | 0.388 |
| Confidence threshold = 0.99 | 3.76 | 6.01 | 0.389 | 0.399 |

Table 8: Stability across different point trackers. We show results with the best confidence threshold (conf.thr.) for each point-tracker.

| Method | MAE ↓ | RMSE ↓ | ArcFace ↑ | Met3R ↓ |
|---|---|---|---|---|
| Cotracker3 (Karaev et al., 2024a) (conf.thr. = 0.98) | 3.63 | 5.81 | 0.390 | 0.388 |
| Alltracker (Harley et al., 2025) (conf.thr. = 0.8) | 3.60 | 5.75 | 0.392 | 0.382 |

et al., 2015) dataset. For dataset preprocessing, we sampled 200 random points on each object and obtained multi-view correspondences by rendering these points onto a fixed set of 20 orbital viewpoints. We additionally use object masks to train our model. Overall, we obtain approximately 1000 object samples for training.

We apply the same architecture and training procedure as described in the main experiments, with the following modifications for the chairs domain. For regularization of the canonical space, we use keypoints obtained by KeypointNet (You et al., 2020) instead of Mediapipe landmarks used for heads. For augmentation, we use random rotation augmentation on [0, 90, 180, 270] degrees to reduce overfitting. The model is trained for 25K steps, fewer than in the head experiments due to the smaller dataset and the synthetic nature of the data.

We demonstrate the learned canonical coordinates (see Fig.12) and dense warping (see Fig. 13). While the results can be further improved with more extensive experiments, these initial findings suggest that our approach holds promise when applied to other domains.

## F  HEAD POSE ESTIMATION

We demonstrate head pose estimation as an additional application of our algorithm. We design a head pose regression model that accepts a monocular image represented by its DenseMarks embeddings as input and produces its 3-DoF head pose as output. By following an established head pose estimation benchmark (Hempel et al., 2022; Ruiz et al., 2018; Zhang et al., 2020), we train our head pose estimation model on the 300W-LP dataset (Zhu et al., 2016) and evaluate it on the AFLW2000-3D dataset (Zhu et al., 2016).

In the table below, we demonstrate that, compared to raw RGB input, using DenseMarks embeddings as input results in a $3\times$ lower angular error for a standard lightweight regressor MobileNet (Howard et al., 2017). We further observe that the head pose can be estimated from DenseMarks input by reducing the regressor to just a single fully-connected layer, while this does not seem to be possible from RGB input.

For all experiments, training was performed using MAE loss, averaged over three angles, and a learning rate of 1e-4. To adapt to more varying cropping in AFLW2000-3D than in 300W-LP, random crop and zoom were added as augmentations in all experiments. Positional encoding (sine/cosine-based) is added as an extra input channel in all experiments. 90% of 300W-LP were used for training and 10% for validation. As a quality reference, one of the most recent methods (Hempel et al., 2022) achieves Avg MAE of $3.47°$ on AFLW2000-3D after training on the full 300W-LP.

## G  SYNTHETIC STRESS TESTS

To evaluate the robustness of our method to various image perturbations, we conduct synthetic stress tests on the LPFF (Wu et al., 2023) dataset. We apply four types of augmentations to test the stability

Table 9: Stability of Densemarks to the errors of the off-the-shelf sparse landmarks predictor (left) and segmentation masks (right).

| Method | MAE ↓ | RMSE ↓ | ArcFace ↑ | Met3R ↓ |
|---|---|---|---|---|
| $\sigma = 16$ | 4.33 | 6.92 | 0.387 | 0.386 |
| $\sigma = 8$ | 4.15 | 6.63 | 0.384 | 0.392 |
| $\sigma = 4$ | 3.84 | 6.14 | 0.380 | 0.389 |
| Ours ($\sigma = 0$) | 3.85 | 6.17 | 0.388 | 0.384 |

| Method | MAE ↓ | RMSE ↓ | ArcFace ↑ | Met3R ↓ |
|---|---|---|---|---|
| Dilation ×16 | 3.99 | 6.37 | 0.381 | 0.382 |
| Dilation ×8 | 3.84 | 6.15 | 0.389 | 0.383 |
| Dilation ×4 | 3.80 | 6.08 | 0.388 | 0.382 |
| Ours (no dilation) | 3.85 | 6.17 | 0.388 | 0.384 |

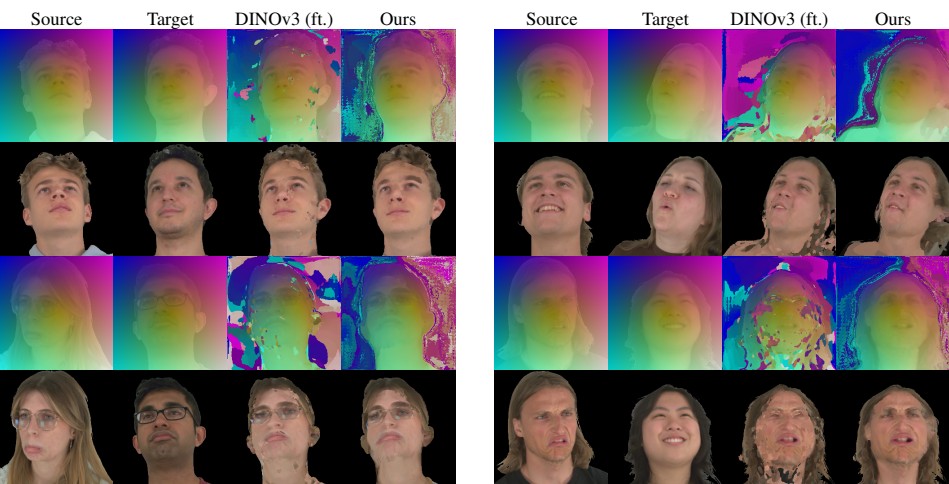

Figure 11: Dense warping, even rows: comparison of our method with DINOv3 fine-tuned. Odd rows: mapping of meshgrid-like coordinates, blended with RGB. Canonical space boosts geometric consistency, indicated by smoother warping.

of correspondences under challenging conditions: motion blur, illumination changes, color shifts, and synthetic occlusions.

We used 300 images from the LPFF dataset, containing diverse poses and appearances. We applied four types of augmentations: (1) motion blur with an intensity range from 15 px to 25 px; (2) illumination changes with brightness and contrast adjustments in the range of -40% to 40%; (3) color shifts with RGB shift in the range of -30 to 30; (4) synthetic occlusions covering the image with uniformly colored rectangles on average by 10%.

For each augmentation type, we apply the augmentation 8 times to each source image and find the nearest neighbors in the original source image. We then fit a Gaussian distribution on the resulting nearest neighbor locations compared with the center in the corresponding pixel. We report three metrics: (1) the mean absolute error (MAE) in pixels; (2) the empirical standard deviation $\sigma_{std}$ of the set of estimates in pixels; (3) the entropy calculated using the obtained standard deviation by the formula for the entropy of a fitted isotropic Gaussian distribution: $H = \ln(2\pi e \sigma_{std}^2)$. Among all input image manipulations, our method seems to be the most sensitive to synthetic occlusion (see Table 11). We find that natural, given that such occlusions were not present in the training data. Nevertheless, even under synthetic occlusion, the standard deviation of the estimated locations remains relatively small (less than 6 pixels). We assume that the entropy can be reduced by incorporating these augmentations into the training process. We provide visualizations of dense warping under lightning change in Fig. 14.

## H ADDITIONAL FAILURE CASES FOR WARPING

We have manually selected challenging images from the LPFF dataset (Wu et al., 2023) with voluminous hairstyles, hats, and masks, and obtained dense warping using correspondences estimated by our method (see Fig. 15). We have not identified significant failure cases related to skin tone differences. Accessories and long hair may constitute a failure case, as the correspondences on them may not always be geometrically consistent, resulting in unnatural distortions on the resulting warps. We

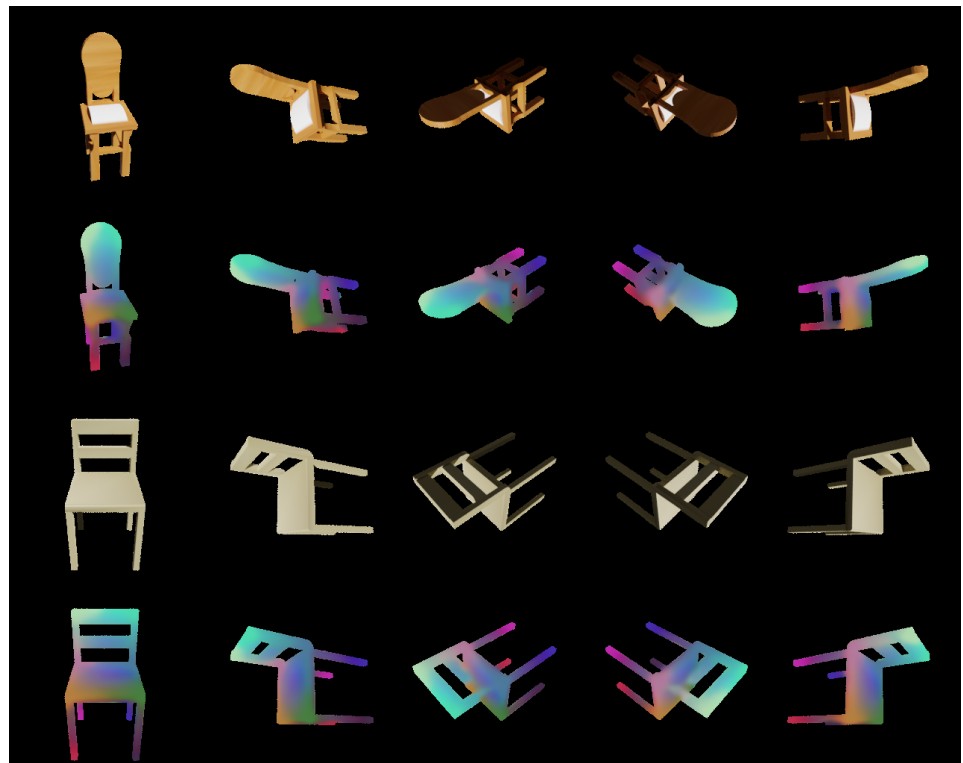

Figure 12: Canonical coordinates for chairs. We learn the canonical space on chairs and visualize RGB and predicted coordinates.

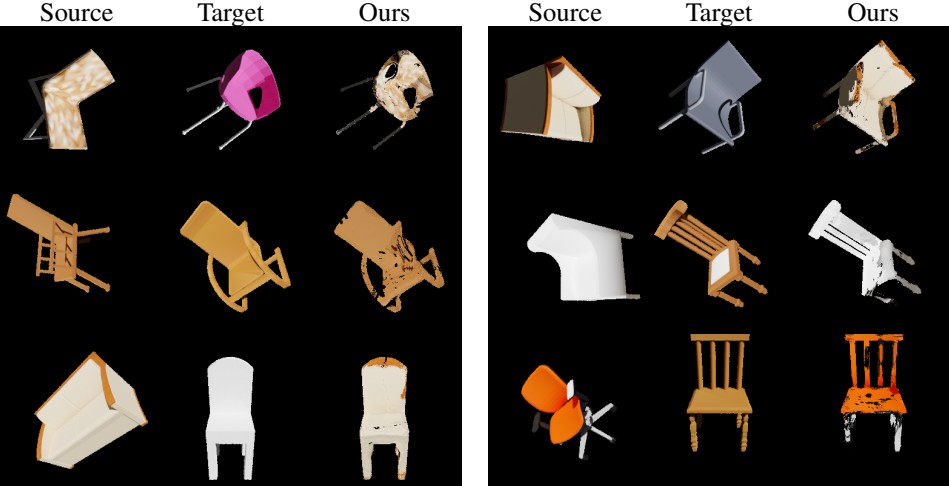

Figure 13: Dense warping. Here, we copy pixels from source to target based on the target→source nearest neighbors search in the space of embeddings, predicted by our model.

believe that our method can potentially benefit from the inclusion of even more diverse datasets in training to alleviate problems in challenging regions.

## I    COMPARISON WITH DENSE FACE METHODS

We did not consider face 3DMM-related baselines (Feng et al., 2018) since they do not provide semantics outside of the face region and cannot be directly used for hair and accessories. For com-

Table 10: Head pose estimation results on 300W-LP validation set and AFLW2000-3D test set.

| | 300W-LP (validation) | | | | AFLW2000-3D | | | |
|---|---|---|---|---|---|---|---|---|
| | Yaw MAE ↓ | Pitch MAE ↓ | Roll MAE ↓ | Avg MAE ↓ | Yaw MAE ↓ | Pitch MAE ↓ | Roll MAE ↓ | Avg MAE ↓ |
| RGB → 1 FC layer | 50.59° | 9.83° | 11.38° | 23.93° | 30.92° | 18.77° | 17.91° | 22.53° |
| DenseMarks → 1 FC layer | 4.30° | 5.45° | 6.72° | 5.49° | 15.00° | 13.26° | 12.58° | 13.61° |
| RGB → MobileNet | 11.60° | 4.06° | 5.61° | 7.09° | 27.74° | 10.57° | 16.37° | 18.22° |
| DenseMarks → MobileNet | 1.81° | 2.75° | 2.40° | 2.32° | 4.13° | 7.77° | 6.03° | 5.97° |

Table 11: Synthetic stress test results on LPFF dataset.

| Method | MAE ↓ | Std $\sigma_{std}$ ↓ | Entropy ↓ |
|---|---|---|---|
| Ours (motion blur) | 2.87 | 3.57 | 5.38 |
| Ours (illumination change) | 1.44 | 2.33 | 4.53 |
| Ours (color shift) | 0.53 | 0.77 | 2.30 |
| Ours (occlusions) | 3.44 | 5.89 | 6.38 |
| Ours (all) | 2.1 | 4.26 | 5.72 |

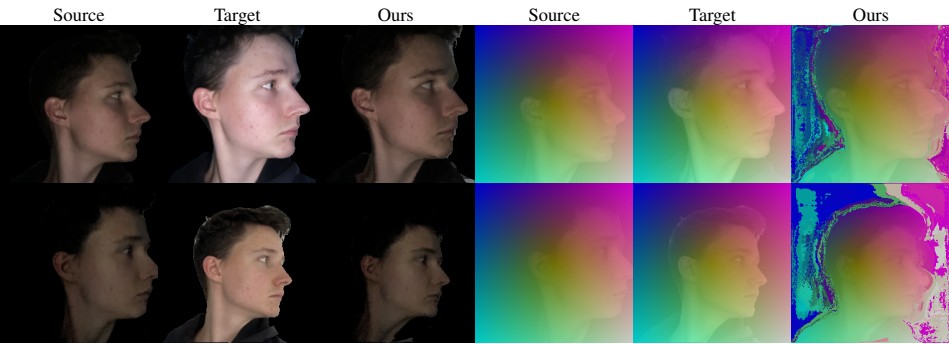

Figure 14: Predicted canonical coordinates are robust to lighting changes. Ours: result of target→source dense warping.

pleteness of the evaluation, we compare with the PRNet baseline (Feng et al., 2018), which predicts coordinates of the Basel Face Model (Paysan et al., 2009). Since it only covers the face region, we develop a heuristic that allows us to compare face 3DMM-related methods outside the face region as well, for the sake of fair comparison. Namely, for every pixel $(i, j)$ in the face region, we leverage $[u, v, 0]$ as embedding, and outside of it, we use $[u_{near}, v_{near}, dist]$, where $(u_{near}, v_{near})$ are UV coordinates of the nearest pixel covered by PRNet, and $dist$ stands for the normalised distance between the pixel $(i, j)$ and the nearest covered pixel. This way, we cover the whole head region with embeddings.

We additionally present values of the PCK measure to quantify the accuracy of correspondences computed across distinct subjects, since ArcFace and Met3R lack true correspondence accuracy. For this, we manually annotated landmarks for 30 random subjects in the face regions (ears, eyebrows, hairline – 15 unique points) on the LPFF dataset (Wu et al., 2023), which includes a wide range of extreme poses/angles and diverse appearances. This gives 400 unique pairs for evaluation.

As we show in Table 12, PRNet performs worse over the same person's correspondence benchmarks compared to our strongest baseline (DINOv3 fine-tuned) and Ours. We observe strong performance of PRNet over cross-person correspondences. Both ArcFace and Met3r seem to strongly rely on the face region, as it constrains the most distinctive features. Additionally, Met3r measures view-consistency, which can be higher if the object is simpler. Nevertheless, our model demonstrates superior performance with respect to the PCK-based benchmarks, which implies more accurate face/head understanding. A potential improvement direction for DenseMarks would be to additionally leverage data with only the face region observed, similar to the data used for training PRNet.

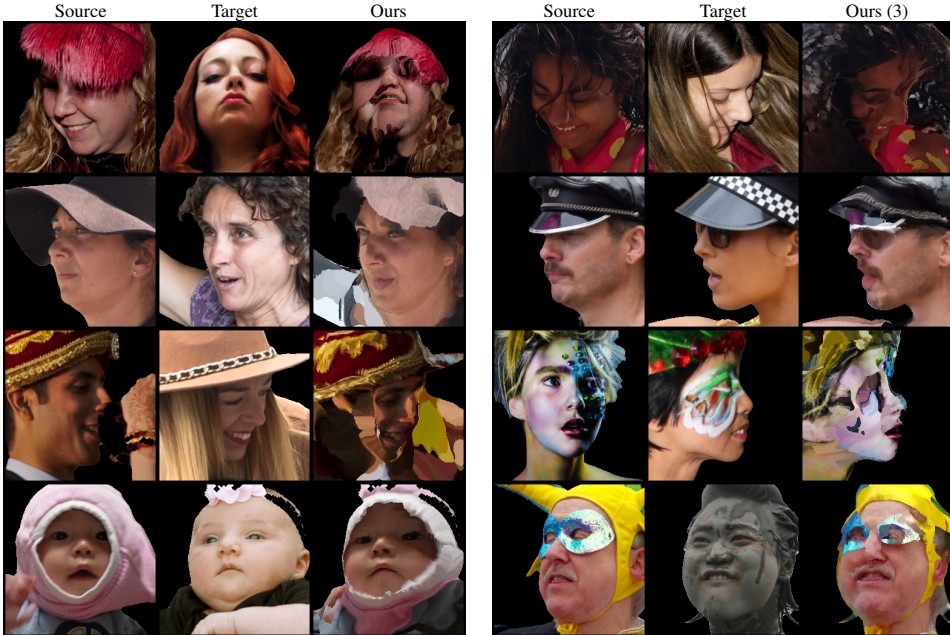

Figure 15: Dense warping. Here, we copy pixels from source to target based on the target→source nearest neighbors search in the space of embeddings, predicted by each model. Our method may struggle to warp long hair, hats, and face masks, resulting in unnatural distortions.

Table 12: Comparison with face 3DMM-related baseline.

|  | Same-person | | | Cross-person | | |
|---|---|---|---|---|---|---|
|  | MAE ↓ | RMSE ↓ | PCK@r=0.05 ↑ | ArcFace ↑ | Met3R ↓ | PCK@r=0.1 ↑ |
| PRNet | 7.33 | 11.70 | 0.72 | **0.462** | **0.365** | 0.76 |
| DINOv3 fine-tuned | 6.35 | 10.20 | 0.85 | 0.348 | 0.455 | 0.69 |
| Ours | **3.68** | **5.90** | **0.90** | 0.384 | 0.388 | **0.85** |

## J VISUALIZATION OF POINT TRACKS

We show visual examples of the generated point tracks in Fig. 16. We found that on our training dataset, CelebV-HQ, point tracks are mostly accurate. We performed a manual examination for a subset of 100 videos and found that approximately 9% of the tracks are incorrectly predicted as invisible and around 3 % of the tracks have visually noticeable drift. Most of the missing tracks originate from comparatively less textured surfaces such as hair, clothes, or headwear. On our training dataset, Celeb-VHQ, tracks are predominantly accurate, even on these challenging surfaces. This reliability is in part due to the high resolution of the head region in the videos and the typically smooth head motion (e.g., interview-style and film shooting data), which allows the point tracker to produce reliable correspondences.

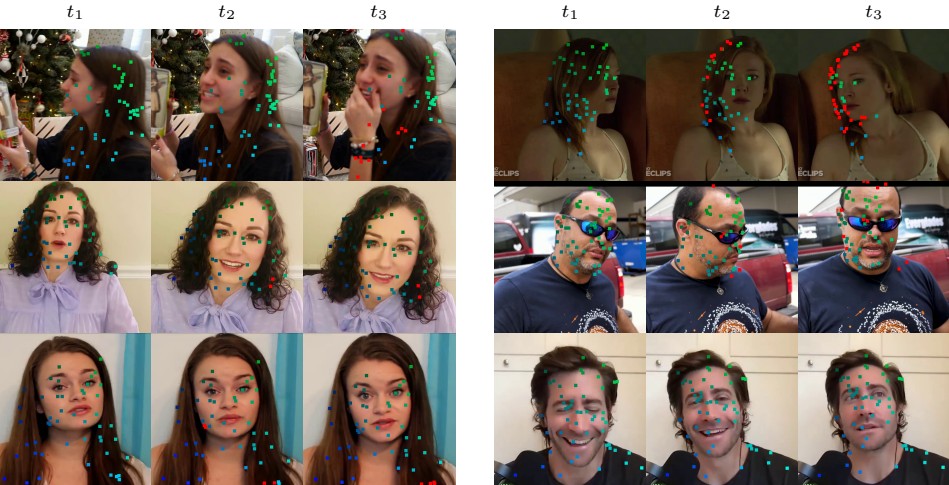

Figure 16: Point-track visualizations on CelebV-HQ dataset; each row shows two videos with three frames each. Starting from $t_1$ tracks are tracked through $t_3$. Missing tracks are indicated by the red color.

