# OpenReview forum: "Densemarks: Learning Canonical Embeddings for Human Heads Images via Point Tracks"
_ICLR.cc/2026/Conference — ICLR 2026 Poster_

### Official Review · Reviewer_zpxw · 2025-10-30

**Soundness:** 2
**Presentation:** 3
**Contribution:** 2
**Rating:** 2
**Confidence:** 4

**Summary:**

The article proposes DenseMarks, a per-pixel embedding of head images into a 3D canonical unit cube. A ViT-based embedder maps each pixel to a 3D coordinate. These coordinates learned a volumetric latent grid that is smoothed with a 3D Gaussian to impose spatial continuity and interpretability. Training uses siamese supervision from automatically extracted point correspondences (CoTracker3) over in-the-wild talking-head videos, plus auxiliary landmark and segmentation losses to anchor semantics (300W landmarks via Mediapipe; parsing via FaRL/SegFormer). The embedding is intended to yield dense, semantics-aware correspondences robust to pose/occlusion, including hair and accessories. Experiments include qualitative point querying, dense warping comparisons to DINOv3, Sapiens, Diffusion Hyperfeatures, Fit3D, quantitative evaluation with ArcFace/Met3R on Nersemble, an application to monocular head tracking, and a small stereo reconstruction demo.

**Strengths:**

A compact 3D canonicalization with an explicit volumetric latent grid and smoothing makes the embedding queryable and visually interpretable. The design aims to cover the entire head (incl. hair/accessories) rather than just landmarkable skin.

Avoids expensive 3D ground truth by leveraging point tracks from off-the-shelf trackers (CoTracker3) on large talking-head corpora. It integrates landmark and parsing constraints to stabilize semantics.

Shows tangible improvement when augmenting a parametric 3DMM tracker (VHAP + FLAME) with a DenseMarks photometric term, especially at extreme poses/occlusions, indicating practical usefulness in monocular tracking.

Despite using 3D coordinates as the embedding key, it seems to produce cleaner dense warps than high-dimensional VFM features (DINOv3, Sapiens, DHFeats, Fit3D) in qualitative demos. The ArcFace/Met3R table suggests better same-person geometric consistency.

**Weaknesses:**

The comparisons target generic dense features (DINOv3/Sapiens/DHFeats/Fit3D) rather than head-specific dense correspondence baselines (e.g., UV/texture-coordinate regressors for faces, functional-map-based human correspondences, dense face/ear/hair parsing and matching). Without task-matched baselines, SOTA claims are difficult to substantiate.

Training depends on CoTracker3 tracks, Mediapipe landmarks, and FaRL/SegFormer parsing, etc. These are all off-the-shelf outputs with their own biases/failure modes. The paper does not quantify noise tolerance or show that DenseMarks isn’t just using these tools’ error modes (e.g., failure on occlusions, hairstyles, accessories).

ArcFace similarity and Met3R measure identity or multi-view image consistency is not explicit correspondence accuracy. The paper lacks standard geometric metrics such as 2D endpoint error (on projected mesh vertices), PCK/NME on annotated parts, or reprojection errors with GT geometry. The Nersemble evaluation uses GS2Mesh meshes as a proxy, but the reported table aggregates identity/consistency rather than direct correspondence accuracy distributions.

Current evidence is largely qualitative. Claims of robustness to occlusion/pose/appearance would benefit from controlled experiments (synthetic occlusions, blur, lighting/colour shifts) and cross-dataset generalization beyond CelebV-HQ/Nersemble.

The paper introduces a 3D latent grid E with Gaussian smoothing ($\sigma$) but does not provide an ablation study for grid resolution ($N^d$), $\sigma$, or memory/speed trade-offs.

Only a qualitative ablation removes landmark or segmentation losses. There’s no quantitative study of $λ_{lmks}$ / $λ_{segm}$ (Eqn. 1), or of how correspondence quality moves with/without each term, nor an analysis of the negative sampling strategy in the contrastive term.

**Questions:**

Missing results on 2D EPE (in pixels) and PCK@$\tau$ on projected mesh vertices for same-person pairs (Nersemble), and part-wise accuracy (ears, hairline, eyebrows) using manual landmarks? This would complement ArcFace/Met3R with true correspondence accuracy.

Missing comparison to dense face UV / canonical correspondence [1, 2] (e.g., PRNet-style [3] face UV maps), CSE/functional-map-style human correspondences, and any dense face parsing + nearest-neighbour baselines. Also, add VFM+learned projection heads trained with the same track supervision to separate the value of canonicalization from pure feature learning.

[1] Alp Guler, R., Trigeorgis, G., Antonakos, E., Snape, P., Zafeiriou, S. and Kokkinos, I., 2017. Densereg: Fully convolutional dense shape regression in-the-wild. In Proceedings of the IEEE conference on computer vision and pattern recognition (pp. 6799-6808).

[2] Guo, J., Zhu, X., Yang, Y., Yang, F., Lei, Z. and Li, S.Z., 2020, August. Towards fast, accurate and stable 3d dense face alignment. In European Conference on Computer Vision (pp. 152-168). Cham: Springer International Publishing.

[3] Feng, Y., Wu, F., Shao, X., Wang, Y. and Zhou, X., 2018. Joint 3d face reconstruction and dense alignment with position map regression network. In Proceedings of the European conference on computer vision (ECCV) (pp. 534-551).

Provide a grid of ($N^d$, $\sigma$) vs accuracy/warp smoothness and memory, test no smoothing / different smoothers. This will validate that the canonical bottleneck (E + smoothing) is the right bias.

Quantify how tracker noise and parsing/landmark errors affect training (e.g., by injecting synthetic correspondence noise, or filtering by CoTracker confidence). Show stability across different point-trackers.

Evaluate cross-dataset (e.g., Nersemble images for testing only, or other head datasets) and stress tests (pose extremes, occlusions, motion blur, illumination changes). Report calibration/uncertainty of matches (entropy of nearest-neighbours’ distributions) under shift.
Provide quantitative curves for $λ_{lmks}$, $λ_{segm}$, and study negative-pair sampling in the contrastive loss (within-frame vs cross-frame; hard-negative mining).

Show representative failure cases (e.g., voluminous hairstyles, hats, masks, skin tones) and per-attribute breakdown to understand biases inherited from CelebV-HQ and the pseudo-label tools.

---

> ### Author Response · Authors · 2025-11-21
> **Official response to the reviewer zpxw**
>
> Thank you for the thoughtful review and many useful suggestions. Below, we address the questions and concerns you raised.
>
> **“Missing results on 2D EPE (in pixels) and PCK@ on projected mesh vertices for same-person pairs (Nersemble), and part-wise accuracy (ears, hairline, eyebrows) using manual landmarks? This would complement ArcFace/Met3R with true correspondence accuracy.”**
>
> We present the 2D EPE measure in Table 1 of the main text (column “Same Person”, MAE/RMSE). We measure ArcFace / Met3R only for the cross-person setup since we lack GT correspondences. For a single-person setup, we obtain them from gs2mesh. We will add more clarifications in the revised version of the paper.
>
> As suggested, we add PCK for geometric correspondences in addition to the already provided MAE / RMSE. The latter is equivalent to NME up to the normalisation constant.
>
> We additionally present values of the PCK measure to quantify the accuracy of correspondences computed across distinct subjects. For this, we manually annotated landmarks for 30 random subjects in the proposed regions (ears, eyebrows, hairline -- 15 unique points) on the LPFF dataset [1], which includes a wide range of extreme poses/angles and diverse appearances. This gives ~400 unique pairs for evaluation. We show the results on LPFF below.
>
> |    Method    |  MAE ↓| RMSE ↓ | PCK@r=0.05 ↑  | ArcFace↑  | Met3r ↓ | Landmarks PCK@r=0.05 ↑  | Landmarks PCK@r=0.1↑ |
> |--------|---|---|---|---|---|---|---|
> DINOv3           | 7.60   |12.69 | 0.72  | 0.266 |0.460 | 0.43 | 0.71 |
> Fit3D               | 12.75 | 21.83| 0.57  |0.236 | 0.558 | 0.20 | 0.43 |
> Hyperfeatures | 8.26   |13.29 | 0.72  | 0.329 | 0.454| 0.54 | 0.71 |
> Sapiens           |14.88 | 24.12 | 0.56  |0.167 | 0.595| 0.04| 0.10 |
> Ours                | 3.68  | 5.90   | 0.90  | 0.384 | 0.388| 0.64 | 0.85 |
>
>
>
> [1] Wu, Yiqian, et al. "LPFF: A portrait dataset for face generators across large poses." Proceedings of the IEEE/CVF International Conference on Computer Vision. 2023.
>
>
> **More face-centric baselines. "The comparisons target generic dense features (DINOv3/Sapiens/DHFeats/Fit3D) rather than head-specific dense correspondence baselines (e.g., UV/texture-coordinate regressors for faces, functional-map-based human correspondences, dense face/ear/hair parsing and matching)"**
>
> We kindly ask you to review our response to all reviewers, section **"More face-centric baselines"**.
>
> Additionally, we include suggested metrics and present the results below.
>
> |    Method    |  MAE ↓| RMSE↓  | PCK@r=0.05↑  | ArcFace↑  | Met3r ↓| Landmarks PCK@r=0.05↑  | Landmarks PCK@r=0.1↑ |
> |--------|---|---|---|---|---|---|---|
> FaRL | 21.38 | 34.41 | 0.46 | 0.166 | 0.632 | 0.11 | 0.21 |
> CSE  | 11.22 | 17.92 | 0.55 | 0.359 | 0.490 | 0.31 | 0.58 |
> DINOv3 | 7.60 | 12.69 | 0.72 | 0.266 | 0.460 | 0.43 | 0.71 |
> DINOv3 fine-tuned | 6.35 | 10.20 | 0.85 | 0.348 |0.455 | 0.55 | 0.69 |
> Ours  | 3.68 | 5.90 | 0.90 | 0.384 | 0.388 | 0.64 | 0.85 |
>
> We did not originally consider 3DMM-related baselines ([1, 2, 3] from zpxw) because they only cover the face region, and thus can not be used to obtain dense correspondences for hair/neck/accessories. However, we will share a comparison with these baselines for completeness during the discussion period.
>
> **"Also, add VFM+learned projection heads trained with the same track supervision to separate the value of canonicalization from pure feature learning."**
>
> This corresponds to the "DINOv3 fine-tuned" baseline -- see the table above. We will ensure that it is included in the revised version of our work.
>
> **"Training depends on CoTracker3 tracks, Mediapipe landmarks, and FaRL/SegFormer parsing, etc. These are all off-the-shelf outputs with their own biases/failure modes. The paper does not quantify noise tolerance or show that DenseMarks isn’t just using these tools’ error modes (e.g., failure on occlusions, hairstyles, accessories)."**
>
> Regarding the CoTracker results, we kindly ask you to review our response to all reviewers (section **"Stability of the point tracker"**). Regarding Mediapipe and face parsing, we are currently working on this and will provide the results later in the discussion phase.
>
> **"Show representative failure cases (e.g., voluminous hairstyles, hats, masks, skin tones)"**
>
> We have tested the method's limitations on especially challenging videos found on the Internet and included the results in **“failure_cases.mp4”** as part of the revised version of the Supplementary Material. We will provide more results, grouped by categories of a potential quality hindrance.
>
> **Other concerns.**
>
> We are currently working on additional experiments to address the remaining concerns and will provide the results later on in the discussion phase.

---

> > ### Author Response · Authors · 2025-12-03
> > **Official response to the reviewer zpxw, part 2**
> >
> > **“Quantify how parsing/landmark errors affect training.”**
> >
> > * **Landmarks errors.** Similar to the previous experiment, where we added noise to the CoTracker output to simulate its error, here we simulate error in landmark prediction by adding Gaussian noise to the landmarks. In the table below, we ablate over the standard deviation (in pixels) of the added noise.
> >
> > |    Method    |  MAE ↓ | RMSE ↓ | ArcFace ↑ | Met3R ↓ |
> > |--------|---|---|---|---|
> > $\sigma=16$| 4.33 | 6.92 | 0.387 | 0.386 |
> > $\sigma=8$| 4.15 | 6.63 | 0.384 | 0.392 |
> > $\sigma=4$| 3.84 | 6.14 | 0.380 | 0.389 |
> > Ours ($\sigma=0$) | 3.85 | 6.17 | 0.388 | 0.384 |
> >
> > * **Segmentations mask errors.** We simulate those by dilating regions of a randomly selected class, repeated a given number of times with a 3x3 kernel. After the dilation, the selected region not only misrepresents its own segmentation class but also makes other regions undersegmented.
> >
> > |    Method    |  MAE ↓ | RMSE ↓ | ArcFace ↑ | Met3R ↓ |
> > |--------|---|---|---|---|
> > Dilation $\times 16$  | 3.99 | 6.37 | 0.381   | 0.382  |
> > Dilation $\times 8$    | 3.84 | 6.15 | 0.389   | 0.383  |
> > Dilation $\times 4$    | 3.80 | 6.08 |  0.388 | 0.382 |
> > Ours (no dilation)      | 3.85 | 6.17 | 0.388 | 0.384 |
> >
> > Overall, we observe even stronger robustness of the method against errors in landmarks and segmentation masks compared to errors in CoTracker's estimated tracks. Since the method was already generally robust to CoTracker predictions as well (see the section “Stability of the point tracker” in the reply to all authors), this suggests robustness of DenseMarks to noise in pseudo-GT tools used in training.
> >
> > **"Provide quantitative analysis of $\lambda_\textrm{lmks}, \lambda_\textrm{seg}$."**
> >
> > |    Method    |  MAE ↓ | RMSE ↓ | ArcFace ↑ | Met3R ↓|
> > |--------|---|---|---|---|
> > Ours ($\lambda_\textrm{lmks}=50, \lambda_\textrm{seg}=1$) | 3.85 | 6.17 | 0.388 | 0.384 |
> > $\lambda_\textrm{lmks}=10$ | 5.35 | 8.68 | 0.390 | 0.413 |
> > $\lambda_\textrm{lmks}=250$| 3.92 | 6.29 | 0.393 | 0.387 |
> > $\lambda_\textrm{seg}=0$ | 3.68 | 5.86 | 0.353 | 0.420 |
> > $\lambda_\textrm{seg}=0.2$ | 3.72 | 5.95 | 0.400 | 0.374 |
> > $\lambda_\textrm{seg}=5$ | 3.98 | 6.39 | 0.384 | 0.400 |
> >
> > We observe that increasing $\lambda_{lmks}$ does not further improve the metrics, and making it too low results in inferior model performance.
> >
> > Lowering the segmentation loss weight results in slightly better performance over both single-person and cross-person benchmarks. As shown in the Figure in **“point_findings_lower_seg_loss.pdf”** as a part of the revised Supplementary Material, our model exhibits close canonical space structure to the $\lambda_\textrm{seg}=0.2$ baseline.

---

> ### Author Response · Authors · 2025-12-03
> **Official response to the reviewer zpxw, part 3**
>
> **"Provide a grid of (N, $\sigma$) vs accuracy/warp smoothness and memory, test no smoothing / different smoothers."**
>
> * **No smoothing / different smoothers.** We run three additional experiments: 1) no smoothing, 2) uniform smoothing instead of Gaussian smoothing, with the same kernel size (7x7), 3) a 3D bilateral filter that is known to preserve edges better. Since bilateral filter has two hyperparameters ($\sigma_d$ -- std for the spatial difference and $\sigma_r$ -- std for the intensity difference), we keep $\sigma_d = 1.5$ as in Gaussian filter in our method and take the value of $\sigma_r = 0.1$ giving the best performance over the grid of values tried [0.05, 0.1, 0.2, 0.4].
>
> |    Method    |  MAE ↓ | RMSE ↓ | ArcFace ↑ | Met3R ↓
> |--------|---|---|---|---|
> No smoothing | 4.22 | 6.71 |  0.387  | 0.393  |
> Uniform smoothing | **3.83** | **6.14** |  0.383   | **0.381**  |
> Bilateral filter | 3.97 | 6.35 | 0.392 | 0.390 |
> Gaussian smoothing (Ours)  | 3.85 | 6.17 | **0.388** | 0.384 |
>
>
> We observe slightly better performance with Uniform smoothing by MAE, RMSE, and Met3R, and better performance with Gaussian (ours) by ArcFace. Overall, the quantitative evaluation confirms the importance of using a smooth canonical space, and the specific filter to use appears not to be particularly important. Additionally, we show region finding in the file **“no_smoothing_region_search.pdf”** as a part of the revised version of the Supplementary Material. Without smoothing, the model is still able to roughly locate correct regions in the canonical space, but the space is harder to use directly for querying or region finding.
>
> * **Grid of (N, $\sigma$) vs accuracy/warp smoothness and memory.** We evaluate our method using a number of combinations of different $N_d$ (the number of voxels in the voxel grid by each side) and $\sigma$ (the Gaussian smoothing blurriness level). Memory occupied by the voxel grid grows cubically with $N_d$.
>
> |    Method    |  MAE ↓ | RMSE ↓ | ArcFace ↑ | Met3R ↓
> |--------|---|---|---|---|
> $N_d = 64$, $\sigma = 0.5$                        | 3.65 | 5.86 | 0.391 | 0.390 |
> $N_d = 64$, $\sigma = 1.5$                        | 4.06 | 6.51 | **0.392** | 0.385 |
> $N_d = 64$, $\sigma = 4.5$                        | 4.04 | 6.49 | 0.391 | 0.385 |
> $N_d = 128$ (ours), $\sigma = 0.5$            | **3.64** | **5.80** | 0.390 | 0.386 |
> $N_d = 128$ (ours), $\sigma = 1.5$ (ours) | 3.85 | 6.17 | 0.388 | **0.384** |
> $N_d = 128$ (ours), $\sigma = 4.5$            | 3.89 | 6.25 | 0.386 | 0.386 |
>
> We observe a slight improvement in geometric metrics (MAE, RMSE) with a lower sigma value, while the best variant according to cross-person metrics (ArcFace, Met3R) is achieved with $\sigma=1.5$, either with $N_d=64$ for ArcFace or with $N_d=128$ for Met3R (as in our final method). Overall, metrics do not seem to deviate very significantly with the variation of these parameters.
>
> Memory consumption. The method takes approximately 16 GB GPU memory for $N_d = 128$ and 14 GB for $N_d = 64$. Increasing $N_d$ further significantly increases memory requirements to approximately 32 GB. Sigma does not affect memory consumption. We were unable to experiment with $N_d = 256$ due to the constraints on available GPU memory.

---

> > ### Author Response · Authors · 2025-12-03
> > **Official response to the reviewer zpxw, part 4**
> >
> > **"Evaluate cross-dataset and stress tests (pose extremes, occlusions, motion blur, illumination changes). Report calibration/uncertainty of matches under shift."**
> >
> > We used 300 images from the LPFF dataset, containing diverse poses and appearances. We used four augmentations: motion blur (intensity range: 15 px to 25 px), illumination change (brightness and contrast: -40% to 40%), color shifts (RGB shift: -30 to 30), and synthetic occlusions (covering the image with uniformly colored rectangles on average by 10%). We apply each augmentation 8 times and find nearest neighbors on the source image. Then, we fit a Gaussian distribution on the resulting nearest neighbor locations compared with the center in the corresponding pixel. We report the empirical standard deviation $\sigma_{std}$ of the set of estimates in pixels. We additionally calculate entropy using the obtained standard deviation by the formula for the entropy of a fitted isotropic Gaussian distribution: $H = \ln(2 \pi e \sigma_{std} ^ 2)$.
> >
> > |    Method    |  MAE ↓ | Std $\sigma_{std}$ ↓ | Entropy ↓ |
> > |--------|---|---|---|
> > Ours (motion blur) | 2.870 | 3.57 | 5.38 |
> > Ours (lightning change) | 1.444 | 2.33 | 4.53 |
> > Ours (color shift) | 0.53 | 0.77 | 2.30 |
> > Ours (occlusions) | 3.44 |  5.89 | 6.38 |
> > Ours (all)  | 2.1 | 4.26 | 5.72 |
> >
> > Among all input image manipulations, our method seems to be the most sensitive to synthetic occlusion. We find that natural, given that such occlusions were not present in the training data. Nevertheless, even under synthetic occlusion, the standard deviation of the estimated locations remains relatively small (less than 6 pixels). We assume that the entropy can be reduced by incorporating these augmentations into the training process.
> >
> > We provide visualizations of dense warping under image manipulations in **“dense_warping_stress_test.pdf”** as a part of the revised version of the Supplementary Material. There, we also visually demonstrate that our method is robust to the lighting change, motion blur, and color shifts.
> >
> > **"Missing comparison to dense face UV / canonical correspondence."**
> >
> > We did not initially consider face 3DMM-related baselines since they do not provide semantics outside of the face region and cannot be directly used for hair and accessories.
> >
> > For completeness of the evaluation, we add the PRNet baseline [1] proposed by Reviewer zpxw, which predicts $[u, v]$ coordinates of the Basel Face Model (BFM). Since it only covers the face region, we develop a heuristic that allows us to compare face 3DMM-related methods outside the face region as well, for the sake of fair comparison. Namely, for every pixel $(i, j)$ in the face region, we leverage $[u, v, 0]$ as embedding, and outside of it, we use $[u_{near}, v_{near}, dist]$, where $(u_{near}, v_{near})$ are UV coordinates of the nearest pixel covered by PRNet, and $dist$ stands for the normalised distance between the pixel $(i, j)$ and the nearest covered pixel. This way, we cover the whole head region with embeddings. We provide the results below.
> >
> > |    Method    |  MAE ↓| RMSE ↓ | PCK@r=0.05 ↑  | ArcFace↑  | Met3r ↓ | Landmarks PCK@r=0.05 ↑  | Landmarks PCK@r=0.1↑ |
> > |--------|---|---|---|---|---|---|---|
> > PRNet | 7.33 | 11.70 | 0.72 | **0.462** | **0.365** | 0.50 | 0.76 |
> > DINOv3 fine-tuned | 6.35 | 10.20 | 0.85 | 0.348 |0.455 | 0.55 | 0.69 |
> > Ours  | **3.68** | **5.90** | **0.90** | 0.384 | 0.388 | **0.64** | **0.85** |
> >
> > PRNet performs worse over the same person's correspondence benchmarks compared to our strongest baseline (DINOv3 fine-tuned) and Ours. We observe strong performance of PRNet over cross-person correspondences. Both ArcFace and Met3r seem to strongly rely on the face region, as it constrains the most distinctive features. Additionally, Met3r measures view-consistency, which can be higher if the object is simpler. Nevertheless, our model demonstrates superior performance with respect to the PCK-based benchmarks, which implies more accurate face/head understanding. A potential improvement direction for DenseMarks would be to additionally leverage data with only the face region observed, similar to the data used for training PRNet.
> >
> > [1]  Feng, Y., Wu, F., Shao, X., Wang, Y. and Zhou, X., 2018. Joint 3d face reconstruction and dense alignment with position map regression network. In Proceedings of the European conference on computer vision (ECCV) (pp. 534-551).
> >
> > [2] Blanz, Volker, and Thomas Vetter. "A morphable model for the synthesis of 3D faces." Seminal Graphics Papers: Pushing the Boundaries, Volume 2. 2023. 157-164.

---

> ### Author Response · Authors · 2025-12-03
> **Official response to the reviewer zpxw, part 5**
>
> **"Show representative failure cases (e.g., voluminous hairstyles, hats, masks, skin tones) and per-attribute breakdown to understand biases inherited from CelebV-HQ and the pseudo-label tools."**
>
> We have manually selected challenging images from the LPFF dataset with voluminous hairstyles, hats, and masks, and obtained dense warping using correspondences estimated by our method (see **“failure_cases_warping.pdf”** as a part of the revised Supplementary Material). We have not identified significant failure cases related to skin tone differences. Accessories and long hair may constitute a failure case, as the correspondences on them may not always be geometrically consistent, resulting in unnatural distortions on the resulting warps. We believe that our method can potentially benefit from the inclusion of even more diverse datasets in training to alleviate problems in challenging regions.
>
> **“Show stability across different point-trackers.”**
>
> We assess our method’s performance against point track detection and localisation performance by randomly including and perturbing tracks for training (see the section "Stability of the point tracker" in our previous response to all reviewers). Achieving competitive performance of our method is possible by using *any point tracker* yielding 40% or larger point track detection rate, and up to 4 px point track localisation error, w.r.t. the CoTracker3 baseline tracker. Furthermore, a 10% point track detection rate and up to 16 px error still yield better performance than all considered baselines, indicating robustness of our method to pseudo-GT errors. Even though we did not manage to process our whole dataset with other trackers due to the compute constraints and limited time, we are currently working on that and will provide the analysis of performance w.r.t. other trackers in the camera-ready version.
>
> **“Study negative-pair sampling in the contrastive loss (within-frame vs cross-frame; hard-negative mining)”.**
>
> We provide an analysis of various oversampling strategies in our response to Reviewer 99jL. The analysis suggests that oversampling harder examples tends to result in slight improvements in geometric correspondence metrics (MAE, RMSE) but does not improve other metrics (ArcFace, Met3R).
>
> **“Filtering by CoTracker confidence”.**
>
> We kindly ask you to refer to the section “Additional confidence filtering” in our reply to all reviewers.

---

### Official Review · Reviewer_99jL · 2025-10-30

**Soundness:** 3
**Presentation:** 3
**Contribution:** 3
**Rating:** 6
**Confidence:** 4

**Summary:**

The paper proposes DenseMarks, a method to learn dense canonical 3D embeddings for human head images. A ViT network predicts per-pixel 3D locations in a canonical unit cube, trained using point tracks from CoTracker3 on talking head videos with contrastive loss plus landmark and segmentation constraints. The method shows improvements over foundation models in head tracking and correspondence tasks.

**Strengths:**

- **Sound overall pipeline** The approach of using point tracks for supervision combined with a 3D canonical space is reasonable and well-motivated
- **Comprehensive evaluation** Good experimental validation across multiple tasks (monocular tracking, dense warping, point querying, application to downstream models, etc. with strong baselines
- **Interpretable representation** The 3D cube canonical space with semantic structure is simple and queryable
- **Practical improvements** Demonstrates clear improvements on downstream tasks like monocular head tracking with 3DMMs

**Weaknesses:**

- **Pseudo-GT Quality** CoTracker3 is not optimized for heads, yet no analysis of pseudo-GT quality or its impact on training. Critical questions remain: How do tracking errors affect learning? Should low-quality tracks be filtered? How reliable are tracks on challenging regions (hair, accessories)?
- **Limited training data diversity** CelebV-HQ contains interview-style videos with predominantly frontal/near-frontal poses and limited expression variations. Impact on generalization to extreme poses unclear. Only 100 held-out videos from the same distribution used for evaluation
- **Weak contrastive learning setup** Sampling pairs from the interview video means pose/expression/shape are very similar, making the task easier but potentially less robust. The current setup may not learn discriminative features for challenging cases
- **Limited novelty** The method is more an engineering / pipeline work: standard ViT training with contrastive loss and auxiliary constraints. The main contribution is applying existing components (CoTracker + contrastive learning) to heads with a 3D cube representation

**Questions:**

- How does tracking quality from CoTracker impact the learned representation? Can you provide quantitative analysis of pseudo-GT reliability and explore filtering strategies?
- Have you considered hard negative mining or sampling strategies that encourage learning from more diverse poses/expressions within the contrastive framework?
- How does the method generalize to truly in-the-wild videos with extreme poses, fast motion, or severe occlusions beyond the interview-style training data?

---

> ### Author Response · Authors · 2025-11-21
> **Official response to the reviewer 99jL**
>
> Thank you for the thoughtful review. Below, we address the questions and concerns you raised.
>
> **CoTracker stability. "CoTracker3 is not optimized for heads, yet no analysis of pseudo-GT quality or its impact on training. Critical questions remain: How do tracking errors affect learning? Should low-quality tracks be filtered? How reliable are tracks on challenging regions (hair, accessories)? ... Can you provide quantitative analysis of pseudo-GT reliability and explore filtering strategies?"**
>
> We kindly ask you to review our response to all reviewers, section **"Stability of the point tracker"**.
>
> **Sampling strategies. "Have you considered hard negative mining or sampling strategies that encourage learning from more diverse poses/expressions within the contrastive framework?"**
>
> To explore the impact of having larger volumes of challenging examples on training, we conduct two additional experiments promoting harder pairs sampling during training. We find that the hardest videos (according to the contrastive loss value) predominantly fall into two qualitatively distinct categories (as indicated in CelebV-HQ metadata) where subjects either (1) have long hair or (2) exhibit extreme head rotations. We construct new training datasets by (1) sampling videos with longer hair 10x times more often and (2) sampling videos with extreme head rotations 10x times more often. We present the results below.
>
> |    Sampling strategy    |  MAE ↓ | RMSE ↓ | ArcFace ↑ | Met3R ↓
> |--------------------|------|------|---------|-------|
> | Extreme poses oversampling  | 3.72 | 5.94 | 0.385   | 0.387 |
> | Long hair oversampling   | 3.73 | 5.97 | 0.388   | 0.383 |
> | Ours               | 3.85 | 6.17 | 0.388   | 0.384 |
>
> Indeed, oversampling harder examples tends to result in slight improvements in geometric correspondences metrics (MAE, RMSE). We elaborate further on the data diversity by answering the question below.
>
> **Limited training data diversity. "CelebV-HQ contains interview-style videos with predominantly frontal/near-frontal poses and limited expression variations. Impact on generalization to extreme poses unclear. Only 100 held-out videos from the same distribution used for evaluation."**
>
> We agree that the Celeb-VHQ dataset primarily consists of interview-style / film shooting data. However, to the best of our knowledge, it is among the most extensive (35K videos) publicly available human video collections, featuring a wide range of appearances, and containing videos of sufficient length and resolution of the head region. As extremely challenging sequences, such as those featuring backside head views, extreme head rotations, and very rapid motion, are uncommon in this dataset, we expect a quality drop of DenseMarks on those sequences, especially on the monocular tracking benchmark, which requires most of the per-pixel correspondences to be reliable.
>
> For method evaluation in the main paper, we additionally use videos from the Internet to test monocular tracking under challenging conditions, such as face occlusion and strong head rotations.
>
> To test the limitations of the approach, we have additionally collected 5 in-the-wild videos from the Internet featuring exceptionally challenging scenarios. We provide additional results in the video **“failure_cases.mp4”** as part of the revised version of the Supplementary Material. As observed, the quality drop occurs for the data, which is very different from the original distribution, demonstrating the limitations of our approach. We hope that the emergence of larger human video datasets and the rapid progress in point tracking methods will enable more diverse supervision for training approaches similar to ours in the future.
>
> **Weak contrastive learning setup. Sampling pairs from the interview video means pose/expression/shape are very similar, making the task easier but potentially less robust. The current setup may not learn discriminative features for challenging cases.**
>
> Shapes for the same video will indeed be similar, but we augment the data with random shifts, rotations, and scaling to promote greater shape diversity. Additionally, 10% of the Celeb-VHQ data are labeled as large head turns, and it features a versatile collection of emotions. Since we sample random frames from the video, a fraction of the frames will be distant in time within a single video and are also likely to differ in emotion and pose. As demonstrated in the supplementary video, our method outperforms sparse landmarks in challenging scenarios characterized by strong pose variation and occlusions. Nevertheless, as we discuss above, we agree that our approach could still scale to more diverse data with higher pose/shape variations.

---

> > ### Comment · Reviewer_99jL · 2025-11-25
> >
> > Thanks for the rebuttal!
> > Most of the concerns were addressed. The main concern was the training setup and dependence on off-the-shelf models that might be suboptimal for this domain (also pointed out by Reviewer zpxw). Specifically, if CoTracker fails to produce high-quality tracks, this impacts the correctness of the training data. If we assume that it is mitigated through training on interview-style data with little variations in 3D head pose, then generalization of the final model is the main concern. The additional experiments expand this analysis and evaluate the robustness of Co-Tracker.
> > Though partially addressed, the dependence on multiple other models combined with limited training data (I acknowledge, though, that building a large-scale, diverse dataset is a whole other challenge) is the main limitation of the method. Yet, since I originally put a positive score despite the concerns I will keep the rating. I will also encourage the authors to add visual examples of the generated tracks (including fail cases) to the paper.

---

> > > ### Author Response · Authors · 2025-12-03
> > > **Response to the Official Comment by Reviewer 99jL**
> > >
> > > We are grateful to have resolved the majority of the concerns of the Reviewer 99jL. Here, we comment on the remaining concerns and suggestions of 99jL.
> > >
> > > **"The dependence on multiple other models combined with limited training data (...) is the main limitation of the method."**
> > >
> > > We have additionally provided the dependence on errors from other off-the-shelf models used in our work (face landmarks estimator, segmentation network) in our extended reply to the Reviewer zpxw, parts 2 and 3.
> > >
> > > **"I will also encourage the authors to add visual examples of the generated tracks (including fail cases) to the paper."**
> > >
> > > We will make sure to add those to the camera-ready version and highlight both successful point tracking results (constituting the vast majority of cases) and missing/disappearing tracks.

---

### Official Review · Reviewer_DAuy · 2025-10-31

**Soundness:** 4
**Presentation:** 4
**Contribution:** 4
**Rating:** 10
**Confidence:** 5

**Summary:**

This paper proposes DenseMarks, a new learned representation for the human head. The method uses a Vision Transformer to predict a 3D embedding for each pixel of an input image. In addition, the authors collect a dataset of pairwise point matches and employ multi-task learning with facial landmarks and segmentation constraints. The proposed representation can be applied to multiple downstream tasks. Experiments demonstrate state-of-the-art performance in geometry-aware point matching and monocular head tracking with 3D Morphable Models.

**Strengths:**

- **High novelty and insightful contribution:**
The paper introduces DenseMarks, a novel dense representation for human heads that:
(1) enables high-quality dense correspondences across complete head regions, including irregular features such as hair and accessories;
(2) achieves robust tracking under challenging conditions such as severe occlusions; and
(3) produces a structured, interpretable, and smooth canonical latent space that supports further exploration and interaction.

Compared with conventional sparse landmark representations, DenseMarks demonstrates remarkable novelty and conceptual depth. Its design is ingenious and inspiring, introducing a unified dense representation that not only achieves fine-grained head correspondence but also exhibits strong generalization and versatility across diverse downstream tasks such as tracking, reconstruction, and semantic understanding. This elegant formulation presents a foundational and broadly applicable idea, with clear potential for extension beyond head modeling to full-body representations and other related domains.
- **Comprehensive related work:**
The authors provide a thorough and well-organized review of related research, demonstrating a deep understanding of the field and clearly situating their contribution within existing literature.
- **Excellent presentation:** The paper includes clear and informative visualization figures and is well written, clearly structured, and compact.
- **Extensive experiments:** The experimental design is comprehensive and thorough, providing strong evidence for the effectiveness and generality of DenseMarks.
- **Excellent visualizations demonstrating the effectiveness of DenseMarks:** Visualization results such as point querying, semantic region mapping on head images, and dense warping clearly show that DenseMarks captures rich and interpretable head semantics and generalizes well across various head-related tasks, including monocular tracking and stereo reconstruction.

**Weaknesses:**

N/A

**Questions:**

**Questions:**

N/A

**Additional comments:**

It would be beneficial for the author to further explore the application of DenseMarks on additional head-related tasks such as head pose estimation [1], which requires both geometric and semantic understanding of head structures. This would provide stronger evidence supporting the generality and effectiveness of the DenseMarks representation.

[1] Algabri, Redhwan, Ahmed Abdu, and Sungon Lee. "Deep learning and machine learning techniques for head pose estimation: a survey." Artificial Intelligence Review 57.10 (2024): 288.

---

> ### Author Response · Authors · 2025-11-21
> **Official response to the reviewer DAuy**
>
> Thank you for the thoughtful review and for your high appreciation of our work. Below, we address the questions and concerns you raised.
>
> **"It would be beneficial for the author to further explore the application of DenseMarks on additional head-related tasks such as head pose estimation [1], which requires both geometric and semantic understanding of head structures. This would provide stronger evidence supporting the generality and effectiveness of the DenseMarks representation."**
>
> We demonstrate head pose estimation as an additional application of our algorithm. We design a head pose regression model that accepts a monocular image represented by its DenseMarks embeddings as input and produces its 3-DoF head pose as output. By following an established head pose estimation benchmark [1, 2, 3], we train our head pose estimation model on the 300W-LP dataset [4] and evaluate it on the AFLW2000-3D dataset [4].
>
> In the table below, we demonstrate that, compared to raw RGB input, using DenseMarks embeddings as input results in a $3\times$ lower angular error for a standard lightweight regressor (MobileNet [5]). We further observe that the head pose can be estimated from DenseMarks input by reducing the regressor to just a single fully-connected layer, while this does not seem to be possible from RGB input.
>
> For all experiments, training was performed using MAE loss, averaged over three angles, and a learning rate of 1e-4. To adapt to more varying cropping in AFLW2000-3D than in 300W-LP, random crop and zoom were added as augmentations in all experiments. Positional encoding (sine/cosine-based) is added as an extra input channel in all experiments. 90% of 300W-LP were used for training and 10% for validation. As a quality reference, one of the most recent methods [1] achieves Avg MAE of 3.47° on AFLW2000-3D after training on the full 300W-LP.
>
> |                           |           | 300W-LP (validation part) |            |           |
> |---------------------------|:---------:|:---------:|:---------:|:---------:|
> |                           | Yaw MAE ↓ | Pitch MAE ↓               | Roll MAE ↓ | Avg MAE ↓ |
> | RGB --> 1 FC layer        |   50.59°  | 9.83°                     | 11.38°     | 23.93°    |
> | DenseMarks --> 1 FC layer |   4.30°   | 5.45°                     | 6.72°      | 5.49°     |
> |                           |           |                           |            |           |
> | RGB --> MobileNet         |   11.60°  | 4.06°                     | 5.61°      | 7.09°     |
> | DenseMarks --> MobileNet  |   1.81°   | 2.75°                     | 2.40°      | 2.32°     |
>
> |                           |           | AFLW2000-3D |            |           |
> |---------------------------|:---------:|:---------:|:---------:|:---------:|
> |                           | Yaw MAE ↓ | Pitch MAE ↓ | Roll MAE ↓ | Avg MAE ↓ |
> | RGB --> 1 FC layer        |   30.92°  |    18.77°   |   17.91°   |   22.53°  |
> | DenseMarks --> 1 FC layer |   15.00°  |    13.26°   |   12.58°   |   13.61°  |
> |                           |           |             |            |           |
> | RGB --> MobileNet         |   27.74°  |    10.57°   |   16.37°   |   18.22°  |
> | DenseMarks --> MobileNet  |   4.13°   |    7.77°    |    6.03°   |   5.97°   |
>
> For more details on training curves for these experiments, see file **“head_pose_estimation_training_curves.pdf”** in the updated revision of the Supplementary.
>
> [1] Hempel, Thorsten, Ahmed A. Abdelrahman, and Ayoub Al-Hamadi. "6d rotation representation for unconstrained head pose estimation." 2022 IEEE International Conference on Image Processing (ICIP). IEEE, 2022.
>
> [2] Ruiz, Nataniel, Eunji Chong, and James M. Rehg. "Fine-grained head pose estimation without keypoints." Proceedings of the IEEE conference on computer vision and pattern recognition workshops. 2018.
>
> [3] Zhang, Hao, et al. "FDN: Feature decoupling network for head pose estimation." Proceedings of the AAAI conference on artificial intelligence. Vol. 34. No. 07. 2020
>
> [4] Zhu, Xiangyu, et al. "Face alignment across large poses: A 3d solution." Proceedings of the IEEE conference on computer vision and pattern recognition. 2016.
>
> [5] Howard, A.G. et al., 2017. MobileNets: Efficient Convolutional Neural Networks for Mobile Vision Applications. CoRR, abs/1704.04861. Available at arXiv: 1704.04861.

---

### Official Review · Reviewer_F5bW · 2025-11-01

**Soundness:** 3
**Presentation:** 4
**Contribution:** 3
**Rating:** 6
**Confidence:** 4

**Summary:**

This paper introduces DenseMarks, a learned dense representation for human head correspondence and tracking. Unlike traditional landmark- or region-based approaches that are limited to sparse and often skin-only regions, DenseMarks learns a per-pixel embedding space that maps every head pixel into a shared 3D canonical unit cube. The canonical embedding provides an interpretable, queryable, and spatially consistent representation across poses, expressions, and individuals, enabling applications such as dense semantic correspondence, 3D morphable head tracking, and view-consistent reconstruction.

The method leverages a Vision Transformer (ViT) backbone that predicts per-pixel embeddings guided by multiple complementary objectives:
1.a contrastive loss based on point correspondences estimated from talking-head videos,
2.semantic supervision from face landmarks and segmentation maps, and
3.a spatial smoothness constraint via a 3D Gaussian-regularized latent feature cube.

Together, these components yield a structured latent space that is both interpretable and geometrically consistent, capturing the full head region, including challenging elements such as hair and accessories. Experimental results show promising performance in dense geometry-aware matching and monocular head tracking, outperforming existing pretrained vision foundation model (VFM) baselines.

**Strengths:**

1. Conceptually strong and well-motivated reformulation of dense correspondence into a 3D canonical latent space. Also, the multi-task supervision yields a highly structured and interpretable representation. Technically sound training strategy that leverages point-tracking data without requiring explicit 3D ground truth.
2. Demonstrates robust performance across pose and occlusion variations, covering the full head including non-skin regions. This paper is easy to understand, and the video instructions are also very clear.

**Weaknesses:**

Method.
The approach depends on the quality and stability of off-the-shelf 2D point trackers, which may introduce bias or drift in contrastive supervision.

Experiment.
Experimental fairness may be affected since the proposed model is trained specifically on face-centric data, while the compared baselines are typically pretrained on broader, more general datasets. This domain bias could partially account for the observed performance gap.

**Questions:**

Is it possible to extend the construction of this representation to the whole body or even to objects in general?

---

> ### Author Response · Authors · 2025-11-21
> **Official response to the reviewer F5bW**
>
> Thank you for the thoughtful review. Below, we address the questions and concerns you raised.
>
> **"The approach depends on the quality and stability of off-the-shelf 2D point trackers, which may introduce bias or drift in contrastive supervision."**
>
>
> We kindly ask you to review our response to all reviewers, section **"Stability of the point tracker"**.
>
> **"Experimental fairness may be affected since the proposed model is trained specifically on face-centric data, while the compared baselines are typically pretrained on broader, more general
> datasets. This domain bias could partially account for the observed performance gap."**
>
> We kindly ask you to review our response to all reviewers, section **"More face-centric baselines"**.
>
> **"Is it possible to extend the construction of this representation to the whole body or even to objects in general?"**
>
> We see extending our method to the whole human body or generic objects as an interesting future research direction.
>
> Our method relies on two main assumptions: (1) a large dataset with multi-view correspondences for the same object, and (2) shared keypoints across objects.
>
> Human heads satisfy both requirements due to the abundance of video data with slow enough motion and large enough resolution, which allows for enriching it with reliable correspondences annotated by a point tracker and robust sparse landmarks.
>
> In theory, we see no hindrance in applying our approach to full bodies or other objects. Even though full-body motion is generally more rapid, individual body parts occupy smaller space in the frame, and full-body keypoint detectors can be less reliable than for faces. The experimental results in the **"Stability of the point tracker"** section of our response to all reviewers indicate that our method is sufficiently resilient to data noise.
>
> Since we are compute-bound, demonstrating full-body results would be tricky on our side due to the additional work required to process and prepare the data for training, given the time-limited discussion phase. We plan to add experiments on generic objects later in the discussion period, likely starting with synthetic objects such as ShapeNet chairs and regularizing canonical space via keypoints obtained by approaches like KeypointNet [1]. We welcome suggestions that would demonstrate this with minimal data processing effort.
>
> [1] You, Yang, et al. "Keypointnet: A large-scale 3d keypoint dataset aggregated from numerous human annotations." Proceedings of the IEEE/CVF Conference on Computer Vision and Pattern Recognition. 2020.

---

> > ### Author Response · Authors · 2025-12-03
> > **Official response to the reviewer F5bW, part 2**
> >
> > **“Is it possible to extend the construction of this representation to the whole body or even to objects in general?”**
> >
> > We have retrained our approach on a sample category of generic objects — specifically, chairs from ShapeNet [1] — and regularized the canonical space using keypoints obtained by KeypointNet [2]. For the dataset preprocessing, we sampled 200 random points on each object and obtained multi-view correspondences by rendering these points onto a fixed set of 20 orbital viewpoints. We additionally use object masks to train our model. Overall, we obtain approximately 1000 object samples. In training, we apply random rotation augmentation on [0, 90, 180, 270] degrees to reduce overfitting, and train for 25K steps. Results of the learned canonical coordinates and dense warping are shown in **“chairs.pdf”** as a part of the revised Supplementary Material. While the results can be further improved with more extensive experiments, these initial findings suggest that our approach holds promise when applied to other domains.
> >
> > [1] Chang, Angel X., et al. "ShapeNet: An information-rich 3d model repository." arXiv preprint arXiv:1512.03012 (2015).
> > [2] You, Yang, et al. "KeypointNet: A large-scale 3d keypoint dataset aggregated from numerous human annotations." Proceedings of the IEEE/CVF Conference on Computer Vision and Pattern Recognition. 2020.

---

### Author Response · Authors · 2025-11-21
**Official response to all reviewers, part 1**

We thank all the reviewers for taking the time to evaluate our work and appreciate its strengths:

- novelty and design choices: _"conceptually strong and well-motivated reformulation"_ (F5bW), _"design is ingenious and inspiring"_, _"elegant formulation"_ (DAuy)
- training strategy: _"multi-task supervision yields a highly structured and interpretable representation"_, _"technically sound training strategy"_ (F5bW), _"the approach of using point trackers for supervision combined with a 3D canonical space is reasonable and well-motivated"_ (99jL), _"avoids expensive 3D ground truth"_ (zpxw)
- experimental validation: _"experimental design is comprehensive and thorough"_ (DAuy), _"good experimental validation across multiple tasks"_ (99jL)
- practical improvements: _"demonstrates clear improvements"_ (99jL), _"shows tangible improvement"_ (zpxw), _"experimental results show promising performance"_ (F5bW), _"experiments demonstrate state-of-the-art performance"_ (DAuy)
- introduced representation: _"together, these components yield a structured latent space that is both interpretable and geometrically consistent"_ (F5bW), _"compared with conventional sparse landmark representations, DenseMarks demonstrates remarkable novelty and conceptual depth"_ (DAuy), _"the 3D cube canonical space with semantic structure is simple and queryable"_ (99jL)

In this response to all reviewers, we address questions and suggestions raised by more than one reviewer. In personalized replies to the reviewers, we either address questions asked by individual reviewers or refer to this response otherwise.

### Stability of the point tracker.

**Most common errors.** We performed a manual examination for a subset of 100 videos and found that approximately 9\% of the tracks are incorrectly predicted as invisible and around 3% of the tracks have visually noticeable drift. Most of the missing tracks originate from comparatively less textured surfaces such as hair, clothes, or headwear. On our training dataset, CelebV-HQ, tracks are predominantly accurate, even on these challenging surfaces.

**Impact of errors on learned representations.** There are, in general, two sources of errors of CoTracker: (1) missing tracks, (2) wrong position of the tracks. We conduct two additional experiments to evaluate their impact. To provide first results during the discussion phase, we utilize 50% of the training steps, which, in our experience, is usually enough for decent convergence. To simulate missing tracks, we randomly keep a specified ratio of the visible tracks for each pair and re-train our model using these tracks only. The results are presented in the table below.

|    # tracks    |  MAE ↓ | RMSE ↓ | ArcFace ↑ | Met3R ↓
|--------|---|---|---|---|
Keep 10% | 4.52 | 7.30 | 0.376 | 0.391 |
Keep 20% | 4.17 | 6.70 | 0.378 | 0.389 |
Keep 40%  | 3.87 | 6.18 | 0.379 | 0.384 |
Keep 80% | 3.86 | 6.19 | 0.381 | 0.383 |
Ours (100%) | 3.85 | 6.17 | 0.388 | 0.384 |

Even when the model sees a smaller fraction of tracks in a particular video, it still learns from a variety of tracks observed across the dataset. Our findings suggest that missing tracks -- the most common source of CoTracker errors -- have only a mild effect on learned representations, as the model still learns from other videos for which tracks are available.

To assess the influence of inaccuracies on the estimated track locations on our method, we simulate track estimation errors by adding random Gaussian noise with increasingly large deviation (expressed in pixels).

|    Noise level    |  MAE ↓ | RMSE ↓ | ArcFace ↑ | Met3R ↓
|--------|---|---|---|---|
$\sigma=16 $ px | 6.29 | 9.97 | 0.355 | 0.436 |
$\sigma=8$ px | 5.14 | 8.20 | 0.361 | 0.411 |
$\sigma=4$ px | 4.26 | 6.81 | 0.382 | 0.390 |
$\sigma=2$ px | 3.86 | 6.19 | 0.383 | 0.378 |
Ours ($\sigma=0$) | 3.85 | 6.17 | 0.388 | 0.384 |

Our findings suggest that even with mild noise in point tracks, our approach can still learn meaningful representations. Even though stronger noise results in a significant decrease in accuracy, especially compared to omitting tracks, the method still performs on par with the baselines. We conclude that video trackers can be used as pseudo-GT on the human heads video data and our approach is robust to mild noise/error in tracks.

We will conduct additional experiments during the discussion period, further filtering tracks based on the predicted confidence. Nevertheless, we do not expect to get significant gains from it, given the close performance of the model for noise levels 2 and 4. However, since we did not initially save the confidence scores, we need to rerun Cotracker on the entire dataset. Given a limited time for the answer and computational resources, we plan to provide them next week.

---

> ### Author Response · Authors · 2025-11-21
> **Official response to all reviewers, part 2**
>
> ### More face-centric baselines.
>
> We expand our experimental section, adding results for three new head-specific baselines below. We thus complement Sapiens, which is a human-specific baseline already included in the main text.
>
> 1. FaRL [1] (the segmentation network used in the segmentation loss in our method)
>
> 2. CSE [2] (proposed by **zpxw**) is a functional map method with publicly available inference code and pre-trained checkpoints. CSE is an extension of DensePose [3] trained using ground-truth point correspondences with a full-body parametric model (SMPL [4]), which is not a requirement of our method.
>
> 3. We fine-tune DINOv3 on our head-specific training data without introducing the canonical space learned by our method.
>
> Our strongest baseline from the paper (DINOv3) is also shown for reference.
>
>
> |    method    |  MAE ↓ | RMSE↓ | ArcFace ↑ | Met3R ↓ |
> |--------|---|---|---|---|
> FaRL | 21.38 | 34.41 | 0.166 | 0.632 |
> CSE  | 11.22 | 17.92 | 0.359 | 0.490 |
> DINOv3 | 7.60 | 12.69 | 0.266 | 0.460 |
> DINOv3 fine-tuned | 6.35 | 10.20 | 0.348 | 0.455 |
> Ours  | 3.68 | 5.90 | 0.384 | 0.388 |
>
> We present dense warping results in the revised version of the Supplementary, in the new file **“dense_warping_additional_baselines.pdf”**, Fig. 1.
>
> FaRL's features likely exhibit a smaller / more localized receptive field (judging by the dense warping visualization), which can explain their inferior performance. CSE demonstrates closer performance to our strongest baseline DINOv3. Still, it is inferior in geometric correspondences to general-purpose DINOv3, our strongest baseline in the paper.
> Further fine-tuning of DINOv3 on our data boosts geometric correspondences and semantic consistency. We will include results for these methods in the revised evaluation, as well as the DINOv3 fine-tuned model.
>
>
> [1] Zheng, Yinglin, et al. "General facial representation learning in a visual-linguistic manner." Proceedings of the IEEE/CVF conference on computer vision and pattern recognition. 2022.
>
> [2] Neverova, Natalia, et al. "Continuous surface embeddings." Advances in Neural Information Processing Systems 33 (2020): 17258-17270.
>
> [3] Güler, Rıza Alp, Natalia Neverova, and Iasonas Kokkinos. "Densepose: Dense human pose estimation in the wild." Proceedings of the IEEE conference on computer vision and pattern recognition. 2018.
>
> [4] Loper, Matthew, et al. "SMPL: A skinned multi-person linear model." Seminal Graphics Papers: Pushing the Boundaries, Volume 2. 2023. 851-866.

---

> ### Author Response · Authors · 2025-12-03
> **Official response to all reviewers, part 3**
>
> ### Confidence-based filtering of point tracks.
>
> By default, CoTracker filters points by discarding them if their estimated confidence is below a threshold of 0.6, which is also what we relied on. Here, we demonstrate what happens when stricter filtering with a larger threshold is applied, thereby removing more outliers. Results are shown below.
>
> |    Method    |  MAE ↓ | RMSE ↓ | ArcFace ↑ | Met3R ↓ |
> |--------|---|---|---|---|
> Confidence threshold = 0.6 (ours) | 3.85 | 6.17 | 0.388 | 0.384
> Confidence threshold = 0.75 | 3.76 | 6.00 |  0.384 | 0.386
> Confidence threshold = 0.9 | 3.81 | 6.11 |   0.385 | 0.389
> Confidence threshold = 0.98 | 3.63 | 5.81 |  0.390 | 0.388
> Confidence threshold = 0.99 | 3.76 | 6.01 |  0.389 | 0.399
>
> In line with our previous observations (section **"Stability of the point tracker"** in our previous response to all reviewers), increasing the threshold should mildly improve the results, as it removes outliers. With a stronger threshold of 0.75 and 0.9, we obtain performance similar to that of 0.6, which is used in our model. With a very strict threshold of 0.98, the model's performance slightly improves indeed. However, if we increase the threshold further (0.99), too many non-outliers are excessively filtered out, and no further improvement is observed.

---

### Author Response · Authors · 2025-12-03
**Summary of the discussion phase for the ACs**

We are very grateful to all the reviewers for their feedback on our work.

Below, we provide a summary of the major concerns raised by the reviewers and give our respective responses, for ACs to navigate the discussion more easily.
_________________

Reviewers F5bW, 99jL, zpxw: **The quality of DenseMarks depends on the quality and robustness of the tools used to create ground truth labels (point tracker, segmentation network, and face landmarks estimator).**

We provide an analysis of the point tracker quality and the robustness of our method to its errors (noise tolerance, missing point tracks) in the section “Stability of the point tracker” of our reply to all reviewers. Additionally, we provide an analysis of robustness to errors of landmarks/segmentation masks in reply to Reviewer zpxw.

We empirically find that injecting Gaussian noise (standard deviation of up to ±4 px) into our point tracks or retaining only a small subset (10%) of all tracks only results in a minor performance decrease for our method. DenseMarks is also robust to large dilations (48 pixels) in ground truth segmentation masks and strong noise in ground truth facial landmarks (up to ±16 px). At a very high level of noise in point tracks (±16 px), the method still outperforms all the baselines. We conclude that not only the selected point tracker, segmentation network, and the landmark estimator provide reliable inputs to our method, but also our learned representations are robust to noise in these pseudo-GT tools.

The provided analysis suggests the potential of the DenseMarks approach in other domains where obtaining accurate point tracks is challenging, such as capturing full-body human data or highly dynamic and non-rigid objects.
_________________
Reviewers F5bW, zpxw: **The evaluation in the paper is based on baselines trained on diverse, not domain-specific data (e.g. DINOv3), and this could explain the performance gap. Therefore, more baselines trained on human faces should be considered.**

We agree with this concern and thus have evaluated more face-centric baselines — FaRL, CSE, and PRNet (see the section "More face-centric baselines" in our reply to all reviewers and the reply to Reviewer zpxw). We have also fine-tuned our strongest baseline (DINOv3) on our dataset of human head images (see *"DINOv3 fine-tuned"* in our reply to all reviewers). CSE and *DINOv3 fine-tuned* perform on par with other baselines; our DenseMarks still outperforms these approaches.
_________________
Reviewer zpxw: **Evaluation should contain more thorough analysis of hyperparameters ($\lambda_\textrm{lmk}$, $\lambda_\textrm{segm}$, $\sigma$, $N_d$, smoothing strategies, confidence filtering) and add correspondence-based metrics (2D EPE, PCK, etc.).**

We provide the evaluation in our responses to Reviewer zpxw and the other reviewers, and sincerely hope that it addresses all concerns and provides sufficient justification for the design choices in the method. As a part of the analysis, we find that individual metrics (but typically not all) can be slightly improved by selecting a different smoothing strategy, reducing the segmentation loss weight, or adjusting the Gaussian blur strength $\sigma$, while visual inspection does not reveal a significant difference to our initially reported results. Other tweaks did not result in quality improvements, but hopefully provide a better understanding of the design choices and hyperparameter values made in the paper. The results will be included in the main and supplementary texts of the camera-ready version.
_________________
Reviewers 99jL, zpxw: **Failure cases of DenseMarks and its robustness to manipulations in input images.**

We provide:
- monocular tracking on challenging cases in the video “failure_cases.mp4” as a part of the revised version of the Supplementary Material,
- quantitative analysis of the quality under stress tests (pose extremes, occlusions, motion blur, illumination changes) in our reply to Reviewer zpxw,
- dense warping visualization for failure cases analysis, as suggested by Reviewer zpxw, in “dense_warping_stress_test.pdf” as a part of the revised version of the Supplementary Material.
_________________
Reviewers F5bW, DAuy: **Applicability for other tasks, such as head pose estimation, and other domains.**

Initial experiments on the head pose estimation application on standard 300W-LP & AFLW2000-3D benchmarks have been provided as a part of our response to Reviewer DAuy. The experiments show a significant improvement from using DenseMarks embeddings as input for the head pose estimation network. We also demonstrate the applicability of DenseMarks to other domains, using the example of a general object (chairs), as a part of our response to Reviewer F5bW.

_________________

We hope that we have responded to all reviewers' concerns. We thank ACs for taking time to evaluate our paper.

---

### Meta-Review · Area_Chair_Xvcw · 2026-01-06

**Summary:**

- The method relies heavily on CoTracker3, Mediapipe landmarks, and other off-the-shelf tools, which may introduce drift, noise, or bias into training. The papers do not quantify the impact of these errors or explore robustness.

- The approach is mainly an application of existing components (CoTracker + contrastive learning + ViT) to head representations, with few methodological innovations or theoretical insights.

- Current evidence is largely qualitative. Claims of robustness to occlusion/pose/appearance would benefit from controlled experiments.

- Limited training data diversity; impact on generalization to extreme poses unclear.

**Reviewer Concerns:**

The rebuttal includes extensive new experiments and analyses, which largely address the reviewers’ concerns.

**Reviewer Scores:**

zpxw is likely to raise their rating, while other reviewers are likely to remain positive.

---

### Decision · Program_Chairs · 2026-01-26

Accept (Poster)